# Globally widespread and increasing violations of environmental flow envelopes

Vili Virkki[1*#], Elina Alanärä[1#], Miina Porkka[1,2], Lauri Ahopelto[1,3], Tom Gleeson[4,5], Chinchu Mohan[4,6], Lan Wang-Erlandsson[7,8], Martina Flörke[9], Dieter Gerten[10,11], Simon N Gosling[12], Naota Hanasaki[13], Hannes Müller Schmied[14,15], Niko Wanders[16], Matti Kummu[1*]

[1] Water and Development Research Group, Aalto University, Espoo, Finland
[2] Global Economic Dynamics and the Biosphere, Royal Swedish Academy of Sciences, Stockholm, Sweden
[3] Finnish Environment Institute, Helsinki, Finland
[4] Department of Civil Engineering, University of Victoria, Victoria, British Columbia, Canada
[5] School of Earth and Ocean Sciences, University of Victoria, Victoria, British Columbia, Canada
[6] Global Institute for Water Security, University of Saskatchewan, Saskatoon, Saskatchewan, Canada
[7] Stockholm Resilience Centre, Stockholm University, Stockholm, Sweden
[8] Bolin Centre for Climate Research, Stockholm University, Stockholm, Sweden
[9] Institute of Engineering Hydrology and Water Resources Management, Ruhr-University Bochum, 44801, Bochum, Germany
[10] Potsdam Institute for Climate Impact Research (PIK), Member of the Leibniz Association, Potsdam, Germany
[11] Humboldt-Universität zu Berlin, Geography Department, Berlin, Germany
[12] School of Geography, University of Nottingham, Nottingham, NG7 2RD, United Kingdom
[13] National Institute for Environmental Studies, Tsukuba, Japan
[14] Institute of Physical Geography, Goethe University Frankfurt, Frankfurt am Main, Germany
[15] Senckenberg Leibniz Biodiversity and Climate Research Centre (SBiK-F), Frankfurt am Main, Germany
[16] Utrecht University, Department of Physical Geography, Utrecht, The Netherlands

# *equal contribution to the article*

* *Correspondence to*: Vili Virkki (vili.virkki@aalto.fi), Matti Kummu (matti.kummu@aalto.fi)

**Abstract.** Human actions and climate change have drastically altered river flows across the world, resulting in adverse effects on riverine ecosystems. Environmental flows (EFs) have emerged as a prominent tool for safeguarding the riverine ecosystems, but at the global scale, the assessment of EFs is associated with high uncertainty related to the hydrological data and EF methods employed. Here, we present a novel, in-depth global EF assessment using environmental flow envelopes (EFEs). Sub-basin specific EFEs are determined for approximately 4,400 sub-basins at a monthly time resolution, and their derivation considers the methodological uncertainties related with global-scale EF studies. In addition to a lower bound of discharge based on existing EF methods, we introduce an upper bound of discharge in the EFE. This upper bound enables identifying areas where streamflow has substantially increased above natural levels. Further, instead of only showing whether EFs are violated over a time period, we quantify, for the first time, the frequency, severity, and trends of EFE violations during the recent historical period.

Discharge was derived from global hydrological model outputs from the ISIMIP 2b ensemble. We use pre-industrial (1801–1860) quasi-natural discharge together with a suite of hydrological EF methods to estimate the EFEs. We then compare the EFEs to recent historical (1976–2005) discharge to assess the violations of the EFE. These violations most commonly manifest themselves by insufficient streamflow during the low-flow season, with fewer violations during the intermediate-flow season, and only a few violations during the high-flow season. The EFE violations are widespread and occurring in half of the sub-basins of the world during more than 5% of the months between 1976 and 2005, which is double compared to the pre-industrial period. The trends in EFE violations have mainly been increasing, which will likely continue in the future with the projected hydroclimatic changes and increases in anthropogenic water use. Indications of increased upper extreme streamflow through EFE upper bound violations are relatively scarce and dispersed. Although local fine-tuning is necessary for practical applications, and further research on the coupling between quantitative discharge and riverine ecosystem responses at the global scale is required, the EFEs provide a quick and globally robust way of determining environmental flow allocations at the sub-basin scale to inform global research and policies on water resources management.

## 1 Introduction

Human exploitation of rivers is a sensitive balance between benefits gained from water use and adverse Earth system responses. While also enabling the development of societies, rivers upkeep two major regulatory Earth system functions: maintaining the hydrological cycle, and providing habitat for freshwater ecosystems (Gleeson et al., 2020). Nonetheless, they are subject to high anthropogenic pressure – e.g. from flow regulation and damming, excessive water withdrawals, pollution, and land use change (Best, 2019; Kummu et al., 2016). Moreover, human-induced climate change can increase or decrease the seasonal streamflow at different spatial scales (Arnell and Gosling, 2013; Asadieh and Krakauer, 2017; Gudmundsson et al., 2021; Moragoda and Cohen, 2020). The pressure on freshwater ecosystems is only expected to increase in the future due to population growth, agriculture (especially irrigation water use), and projected climate change (Best, 2019; Campbell et al., 2017; Graham et al., 2020; Thompson et al., 2021, Wada and Bierkens, 2014).

Freshwater ecosystems contain nearly 6% of all known species concentrated in 0.8% of Earth's surface (Dudgeon et al., 2006). The riverine parts of freshwater ecosystems have been seriously compromised by human actions: rivers containing 65% of the global discharge are classified to be under moderate to high threat in terms of biodiversity (Vörösmarty et al., 2010), 53% of global rivers have experienced marked changes in fish biodiversity (Su et al., 2021), and 48% of global river reaches are impaired by diminished connectivity (Grill et al., 2019). One of the root causes behind this degradation is the anthropogenic alteration of the natural flow regime of a river – i.e. the magnitude, frequency, duration, timing, and rate of change in flow (Poff et al., 1997). Human actions impact the intra- and interannual variability, which are often considered as parts of the natural flow regime (Richter et al., 2006). These natural streamflow dynamics have already changed in major rivers across the globe (Grill et al., 2015). The flow regime is one of the key factors in defining the integrity of riverine ecosystems, as it

maintains their physical habitat as well as their longitudinal and lateral connectivity (Bunn and Arthington, 2002). Furthermore, aquatic species have evolved within and adapted to the natural flow regime, and alterations to it may facilitate invasive species. Therefore, although riverine ecosystems are extremely complex, the association between flow regime alteration and riverine ecosystem integrity is strong (Poff and Zimmerman, 2010; Rolls et al., 2018).

To safeguard riverine ecosystems, the concept of environmental flows (hereafter EFs) has emerged during the past three decades (Poff and Matthews, 2013). While multiple definitions of EFs exist, the most comprehensive recent definition comes from the Brisbane Declaration 2018 (Arthington et al., 2018), which states that "environmental flows describe the quantity, timing, and quality of freshwater flows and levels necessary to sustain aquatic ecosystems which, in turn, support human cultures, economies, sustainable livelihoods, and well-being." To date, many countries have initiated legislation that would support the establishment of EFs as a concrete means of conserving and restoring riverine ecosystems (Acreman et al., 2014; Arthington et al., 2018; Tickner et al., 2020). In an ideal case, EFs are quantified by assimilating observed hydrological data with local-scale expert knowledge in a collaborative process, resulting in EFs tailored for each unique river (Richter et al., 2006; Poff et al., 2017). Such holistic EF methods include, for example, ELOHA (Poff et al., 2010), DRIFT (King et al., 2003), and PROBFLO (O'Brien et al., 2018).

While the holistic methods available to quantify EFs are comprehensive, the data required to implement them are unavailable at a global scale. Hence, in global studies, the concept of EFs is typically quantified by computing environmental flow requirements (EFRs) based on hydrological EF methods (Pastor et al., 2014). These methods assume that not transgressing EFRs will retain a fair state of riverine ecosystems. Although this proxy relationship is uncertain and varies across spatial and temporal scales (Bunn and Arthington, 2002; Poff and Zimmerman, 2010; Rolls et al., 2018), the hydrological EF methods are often used in global studies as presumptive standards of sustaining riverine ecosystems (Gerten et al., 2020, 2013; Hanasaki et al., 2008; Hoekstra and Mekonnen, 2011; Hogeboom et al., 2020; Jägermeyr et al., 2017; Liu et al., 2021; Pastor et al., 2014, 2019; Steffen et al., 2015). In addition to ecological uncertainty, discharge data used for determining EFRs in global studies are uncertain; runoff and discharge estimated by global hydrological models (GHMs) that are forced with modelled climate from general circulation models (GCMs) tend to be highly dispersed between different GHMs and GCMs (Gädeke et al., 2020; Hattermann et al., 2018; Müller Schmied et al., 2016; Schewe et al., 2014; Veldkamp et al., 2018; Zaherpour et al., 2019). As the underlying hydrological data are generally uncertain, determining EFRs based on them and hydrological EF methods is equally uncertain. Moreover, hydrological EF methods often only set a minimum discharge boundary, disregarding the potentially adverse effects of flows increasing significantly above natural levels – especially in floodplain ecosystems (Hayes et al., 2018; Junk et al., 1989; Schneider et al., 2017; Talbot et al., 2018). Although reviews of EFs have recognised the threat of increased upper extreme flows (Acreman et al., 2014; Poff and Zimmerman, 2010; Richter, 2010), and limiting upper extreme flows has been conceptually proposed (Richter et al., 2012), a global scale methodology to quantify this does not yet exist.

Existing global studies are also limited in their EF violation assessment. Commonly, EFs are treated in global studies as simple,
monthly or annual limits that are either violated or not, lacking quantification of how frequently or how severely these violations manifest themselves (Pastor et al., 2014; Steffen et al., 2015). Some more detailed studies incorporate additional factors, such as the magnitude by which EFs are violated, but do not account for the seasonality of streamflow (Hogeboom et al., 2020; Jägermeyr et al., 2017). Given that particularly low flows are often the most impacted by anthropogenic actions, such as water withdrawals and flow regulation by damming (Döll et al., 2009; Schneider et al., 2017), EF assessments should
be able to separate violations during different flow seasons. Finally, while recent studies have shown that river flows have changed considerably due to direct human actions (Graham et al., 2020; Müller Schmied et al., 2016) and climate change (Gudmundsson et al., 2021; Moragoda and Cohen, 2020) during the past decades, no study has yet assessed the past trends in EF violations. Therefore, new knowledge is required to compose a combined and comprehensive outlook on these three aspects of EF violation.


Here, we present an in-depth global EF assessment by applying a robust, global-scale methodology of environmental flow envelopes (EFEs). Defined at the sub-basin scale in monthly time resolution, the EFE is an envelope of discharge variability, which advances the existing methods in two main ways. First, in order to reduce uncertainties in global EF assessments, the EFE is composed of a number of hydrological EF methods applied to an ensemble of GHM outputs simulated using multiple
GCMs. Secondly, we include a preliminary upper bound in the EFE, aiding in identifying areas where streamflow has increased substantially above the presumed natural levels. In addition to the methodological advances, we present a novel quantification of the seasonal frequency, severity, and trends of EFE violations by comparing recent historical (1976–2005), anthropogenically influenced discharge to pre-industrial (1801–1860) state EFEs.

## 2 Methods and data

Estimating EFE violations was divided into three parts, which are outlined in Fig. 1 and detailed in the following sections. Our method is based on discharge data, which are simulated by four GHMs. Simulating discharge with the GHMs involves modelling the global terrestrial hydrological cycle through process-based equations, as well as forcing the models with observed or modelled climate. For this study, we used modelled climate from four different GCMs, resulting in each GHM providing four distinct data sets of gridded global-scale daily discharge. First, for each distinct discharge data set (i.e. GHM-
GCM combination), we transformed the gridded daily discharge to monthly discharge at the sub-basin scale according to HydroBASINS sub-basin division (Lehner and Grill, 2013). This was done separately for the pre-industrial period (1801–1860) and the recent historical period (1976–2005). Second, we took the pre-industrial monthly discharge for each GHM at the sub-basin scale and estimated EFRs using five hydrological EF methods for four discharge data sets from different GCMs. This totalled 20 EFRs (5 EF methods x 4 GCMs) for each sub-basin and each of the four GHMs. From this EFR distribution,

we drew the median as the GHM-specific EFE lower bound for each sub-basin. Further, we determined the EFE upper bound from the pre-industrial discharge – again, separately for each GHM and sub-basin. Finally, we took the monthly discharge at the sub-basin scale from the recent historical period and compared it to the EFEs. This resulted in EFE violations during months in which the recent historical discharge was not within the EFE. Elaborating our results further, we proceeded to estimate the frequency, severity, and trends of EFE violations.

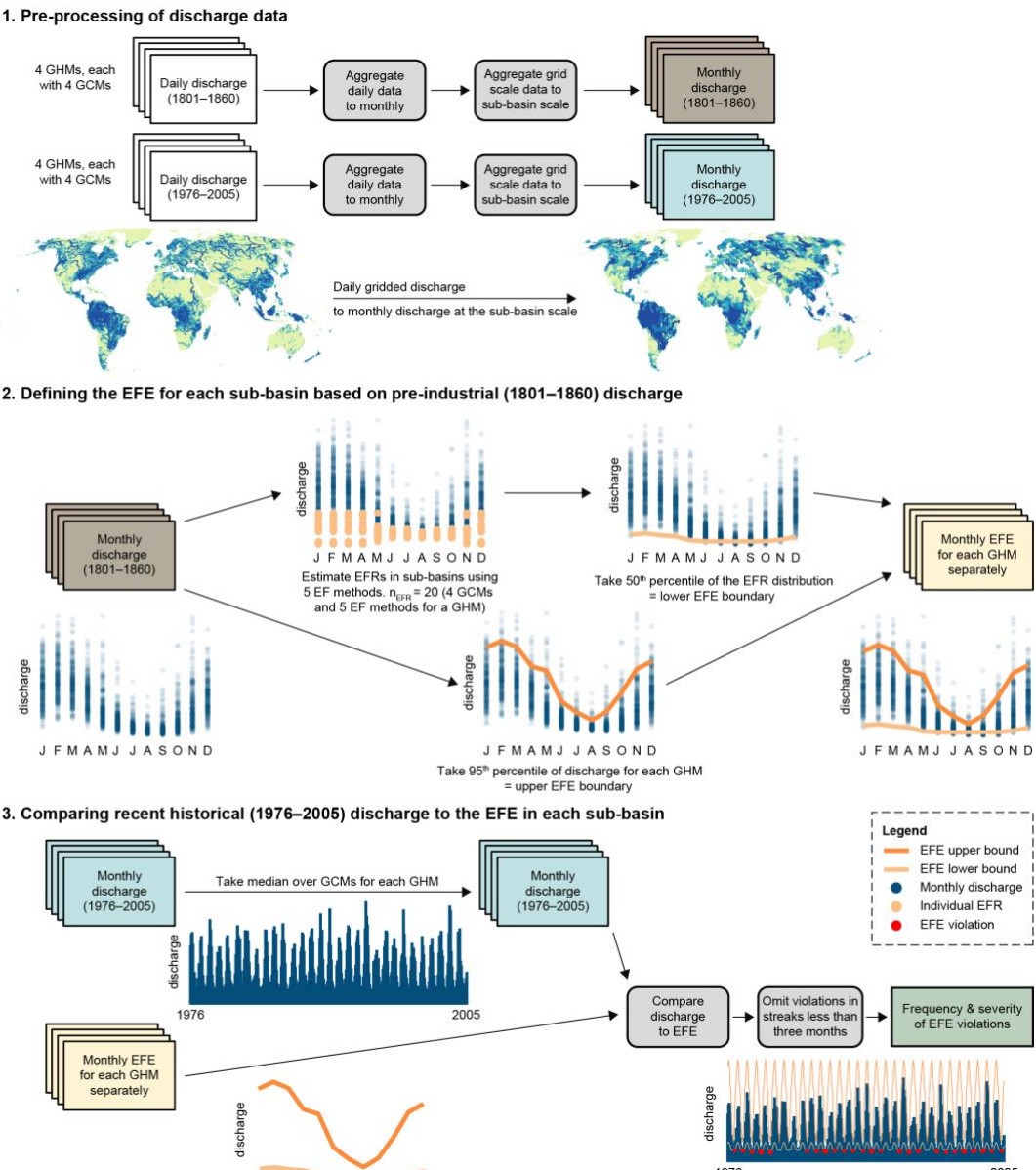


**Figure 1.** The methodological outline of this study: defining the environmental flow envelope (EFE) and estimating the frequency and severity of EFE violations in each sub-basin. GHM stands for global hydrological model, GCM for general circulation model, and EFR for environmental flow requirement.

**2.1 Data**

We used the Inter-Sectoral Impact Model Intercomparison Project (ISIMIP) simulation round 2b outputs of global daily discharge (Frieler et al., 2017; available at: https://esg.pik-potsdam.de). ISIMIP is a community-driven climate impacts modelling initiative that collects and harmonises global model outputs (The Inter-Sectoral Impact Model Intercomparison Project, 2021). To guarantee cross-model consistency regarding the parameterisation of both human and climate factors, the ISIMIP 2b experiments are directed by a protocol (Frieler et al., 2017). Due to the protocol, the outputs of ISIMIP 2b are

comparable between the pre-industrial and recent historical period model runs, as well as between different models.

To decrease the uncertainties related to using single GHMs with single or few GCMs, we chose to use discharge estimates from four different GHMs, namely H08 (Hanasaki et al., 2018), LPJmL (Schaphoff et al., 2018), PCR-GLOBWB (Sutanudjaja et al., 2018), and WaterGAP2 (Müller Schmied et al., 2016). In ISIMIP 2b, each of the GHMs is forced with modelled climate

from four GCMs, namely GFDL-ESM2M (Dunne et al., 2012), HadGEM2-ES (Collins et al., 2011), IPSL-CM5A-LR (Dufresne et al., 2013), and MIROC5 (Watanabe et al., 2010). All of these GCMs were included in our discharge ensemble. The selected GHMs have undergone extensive intercomparison and validation with observed data (see e.g. Gädeke et al., 2020; Zaherpour et al., 2018). Furthermore, the hydrologically best fitting model varies largely based on sub-basin characteristics (Zaherpour et al., 2018), which is why we chose to estimate EFEs for many GHMs instead of pursuing the hydrological best

fit and estimating EFEs for one GHM only. This ensemble approach decreases uncertainty stemming from two separate sources: (1) using more than one GCM within one GHM decreases the GHM parameterisation uncertainty; and (2) using a number of GHMs in an analysis decreases the uncertainty of modelling the hydrological cycle within a single GHM (Döll et al., 2016; Schewe et al., 2014; Sood and Smakhtin, 2015).

The discharge data (for both periods 1801–1860 and 1976–2005) were first temporally aggregated from daily to monthly discharge by calculating the monthly mean of daily values, followed by a spatial aggregation at the sub-basin scale. We used the HydroBASINS sub-basin division, which is a global polygon layer series dividing the world into sub-basins at different scale levels from the lowest detailed level 1 to the highest detailed level 12; we selected the medium detailed level 5 (Lehner and Grill, 2013). Within each level, the geographical areas of sub-basins are relatively equal, and level 5 is the highest level

of detail that can be rasterized into a 0.5-degree resolution grid without an excessive loss of sub-basins that are smaller than a grid cell. In total, 352 out of 4,734 sub-basins were excluded due to being smaller than a grid cell, corresponding to less than 1 million $km^2$. or 1% of the global land area. The average size of the remaining sub-basins was 30,700 $km^2$ and median size 19,600 $km^2$. Minor additional exclusions of five to six sub-basins per GHM were caused by non-overlapping discharge data grids. To aggregate the discharge at the sub-basin scale, we selected the maximum discharge value within the borders of each

sub-basin, assuming that the sub-basin drains out from that location. Hence, we consider this location – and any violation in it – as representative of the whole sub-basin, though the situation may vary in different parts of the sub-basin.

## 2.2 Defining EFEs

We defined the EFEs based on the pre-industrial period (1801–1860). While some flow alteration (e.g. canals) may have already existed by 1860, large-scale human modification of rivers has prevailed mainly after the pre-industrial period. For example, area equipped for irrigation has increased sixfold since 1900 (Siebert et al., 2015), and many of the globally largest dams have been commissioned during the 20th century (Lehner et al., 2011). Therefore, we presumed that this time period is quasi-natural – i.e. near the natural flow regime. Furthermore, reaching back to the pre-industrial time period enabled us to quantify the joint effect of both direct anthropogenic flow alteration and anthropocentric climate change, although explicit separation between these two drivers is not possible with the ISIMIP 2b data used in this study.

Following Pastor et al. (2014), we selected five hydrological EF methods to accommodate for the differences in the methods' definitions of ecosystem water needs. The selected methods include Smakhtin's method (Smakhtin et al., 2004), Tennant's method (Tennant, 1976), Tessmann's method (Tessmann, 1980), the Q90-Q50-method (Pastor et al., 2014), and the variable monthly flow (VMF) method (Pastor et al., 2014). These methods have been validated against *in situ* EFR estimates by Pastor et al. (2014) and Jägermeyr et al. (2017). We opted to use these relatively simplistic hydrological EF methods because more advanced EF quantifications, such as ELOHA-based methods, are limited in global scale applicability due to the high data and resource requirements (Richter et al., 2012). The selected methods are based on simple flow metrics, such as mean annual or monthly flow, and they determine EFRs according to hydrological seasons. All methods distinguish between low-flow and high-flow months, while the Tessmann and VMF methods supplement this with a third class for intermediate-flow months. The equations to compute EFRs according to the selected EF methods are presented in Table 1.

**Table 1.** Descriptions of hydrological EF methods used to calculate environmental flow requirements (EFRs) in this study (adapted from Pastor et al. (2014)). MMF refers to mean monthly flow of each month, MAF to mean annual flow (the mean monthly flow of all months within a year), Q50 and Q90 to flow exceeding 50% and 90% of the flows during the period of interest respectively, and coefHF to high-flow coefficient used in Smakhtin's method.

| Hydrological season | Smakhtin (2004) | Tennant (1976) | Q90-Q50 (Pastor et al., 2014) | Tessmann (1980) | Variable monthly flow (Pastor et al., 2014) |
|---|---|---|---|---|---|
| Low-flow month definition | $MMF \leq MAF$ | $MMF \leq MAF$ | $MMF \leq MAF$ | $MMF \leq 0.4 \times MAF$ | $MMF \leq 0.4 \times MAF$ |
| EFR of low-flow month | $Q90$ | $0.2 \times MAF$ | $Q90$ | $MMF$ | $0.6 \times MMF$ |
| High-flow month definition | $MMF > MAF$ | $MMF > MAF$ | $MMF > MAF$ | $MMF > 0.4 \times MAF$ and $0.4 \times MMF > 0.4 \times MAF$ | $MMF > 0.8 \times MAF$ |
| EFR of high-flow month | $coef_{HF} \times MAF^{(a)}$ | $0.4 \times MAF$ | $Q50$ | $0.4 \times MMF$ | $0.3 \times MMF$ |

| | | | | | |
|---|---|---|---|---|---|
| Intermediate-flow month definition | - | - | - | *MMF > 0.4 x MAF and 0.4 x MMF ≤ 0.4 x MAF* | *MMF > 0.4 x MAF and MMF ≤ 0.8 x MAF* |
| EFR of intermediate-flow month | - | - | - | *0.4 x MAF* | *0.45 x MMF* |

(a) If *Q90 > 0.3 x MAF*, *coef_HF = 0*; if *0.2 x MAF < Q90 ≤ 0.3 x MAF*, *coef_HF = 0.07*; if *0.1 x MAF < Q90 ≤ 0.2 x MAF*, *coef_HF = 0.15*; if *Q90 ≤ 0.1 x MAF*, *coef_HF = 0.2*.

For each GHM, we applied the five EF methods to four discharge data sets simulated using modelled climate from four GCMs, resulting in a monthly distribution of 20 independent EFR estimates per GHM. Before computing EFRs, we removed monthly
outlier discharge further than three standard deviations away from mean monthly discharge. This procedure only removed extremely deviating discharge values, which could greatly distort the computation of EFRs or shift the EFE upper bound very high if left in the data. Similarly for the resulting EFR distribution, EFRs further than three standard deviations away from mean EFR were removed. This way, we avoided skewing the EFR distribution with extreme outliers in pre-industrial data.

From the EFR distribution, we drew the median as the GHM-specific EFE lower bound. This is an ensemble modelling approach, which is often adopted in multi-model studies (Peel and Blöschl, 2011). While the differences in the EF methods provide variability to our ensemble, all of the estimates are hydrologically uncertain. From a set of uncertain estimates, simple ensemble metrics, such as the ensemble mean or median, often provide adequate results at the global level when compared to observed discharge (see e.g. Arsenault et al., 2015; Huang et al., 2017), although individual members of the ensemble may
outperform the simple metrics at the catchment scale (Zaherpour et al., 2018). Selecting the midway EFR excludes the tails of the EFR distribution that potentially consist of unrealistically low or high EFR estimates. These can be caused by, for example, highly deviant discharge provided by certain GCMs, or a poor fit of an individual EF method to the flow regime of a sub-basin. Hence, the GHM-specific EFE lower bound is an ensemble-based consensus estimate between different GCMs and EF methods.


As the EFE upper bound for each GHM, we selected the 95[th] percentile of pre-industrial monthly discharge over all GCMs. While minor flooding can still be beneficial for riverine ecosystems, extreme floods often result in adverse effects (Talbot et al., 2018) and floodplain ecosystems in particular require a distinctive dry period (Hayes et al., 2018; Junk et al., 1989; Schneider et al., 2017). This dry period may be compromised by increased dry season flows – for example, due to hydropower
operation. Other factors that potentially cause increases in flows across all flow seasons include, for example, natural climate variability, anthropocentric climate change, inter-basin water transfers, and land use change. Exceeding the 95[th] percentile of pre-industrial monthly discharge – simulated using modelled climate from all four GCMs – can thereby be considered as a remarkable signal of increased flows, although the underlying drivers vary. Though the mechanism of ecosystem degradation

caused by increased flows is known to exist, no hydro-ecologically grounded quantitative methods have been introduced.

Therefore, we used the 95th percentile as the first step and inspiration towards future methodological advances.

For illustration, a conceptual definition of the EFE is presented in Fig. A1, a comparison between monthly pre-industrial discharge and the EFE lower bound is presented in Fig. A2, and a comparison between EFEs and recent historical discharge in sub-basins in variable flow regimes across the world is presented in Fig. A3.

## 2.3 Evaluating EFE violations

Finally, we compared the recent historical (1976–2005) discharge to the EFE in each sub-basin. The end date of the recent historical period was limited to 2005 by the ISIMIP 2b simulation protocol owing to a lack of reliable estimates regarding, for example, irrigation extent for the years thereafter (Siebert et al., 2015). For each GHM, we calculated a monthly violation ratio between the median discharge over four GCMs and the GHM-specific EFE (Table 2). The violation ratio yields a value

between 0 and 100 if discharge is within the EFE, a negative value if discharge is below the EFE lower bound, and a value over 100 if discharge is above the EFE upper bound. In the few cases where the EFE was unavailable due to no flow during the pre-industrial period, we considered the violation ratio to be zero – i.e. no violation.

**Table 2.** Computing the EFE violation ratio. Q stands for monthly discharge between 1976 and 2005; $EFE_{lower}$ for the EFE lower bound,
and $EFE_{upper}$ for the EFE upper bound.

| Condition | Equation for violation ratio | | Violation ratio |
|---|---|---|---|
| $Q < EFE_{lower}$ | $\dfrac{Q - EFE_{lower}}{EFE_{lower}} \; x \; 100$ | (1) | $< 0$ |
| $EFE_{lower} \leq Q \leq EFE_{upper}$ | $\dfrac{Q - EFE_{lower}}{EFE_{upper} - EFE_{lower}} \; x \; 100$ | (2) | $0 - 100$ (*no EFE violation*) |
| $Q > EFE_{upper}$ | $\left(\dfrac{Q - EFE_{upper}}{EFE_{upper}} + 1\right) x \; 100$ | (3) | $> 100$ |

Throughout the analysis, we excluded time periods during which the EFE is violated for less than three consecutive months. This emphasises persistent flow alterations that are likely to threaten riverine ecosystems beyond individual species (Biggs et al., 2005). Simultaneously, potential one-month outliers in recent historical discharge are eliminated and do not therefore cause

bias to violation metrics. On the other hand, flow alteration events lasting less than three months, such as rapid floods and short-term water withdrawals, are inevitably masked from the results. In addition to results presented in the following section with a minimum three-month sequence of violations, we repeated the analysis with other minimum lengths of the violation streak. The results of this sensitivity analysis are presented in the supplementary material (Fig. S1–S3); shorter minimum violation streaks extend the violations to wider areas, and increasing the minimum violation streak limits the violations to

relatively small regions. Finally, although we defined the EFE for all sub-basins, we excluded sub-basins with extremely low flow from further analysis. If at least three out of four GHMs estimated mean annual flow (the mean monthly flow of all months; MAF) to be less than 10 $m^3$ $s^{-1}$ at the sub-basin outlet, the sub-basin was excluded. During the recent historical period, sub-basins covering 6.5% of the global land area were excluded due to this criterion.

Using equations in Table 2, we determined the violation ratio in each sub-basin for each month in 1976–2005. Considering the four GHMs, this resulted in a total of 1,440 violation ratios for each sub-basin (4 GHMs x 30 years x 12 months). We treated the violation ratios from different GHMs as independent observations of violation since the EFE was defined and evaluated strictly GHM-wise. We then defined two metrics: violation frequency and violation severity. The violation frequency is defined as the fraction of violated months out of all 1,440 months. The violation severity is defined as the unweighted mean of violation

ratios during the violated months, the count of which may vary. These metrics were computed separately for the lower and upper EFE bounds. A numerical example is provided in Fig. A1. In addition to the results presented in the following section, we conducted the analysis for individual GHMs, the results for which are shown in the supplementary material (Fig. S4–S11).

Elaborating the EFE violation patterns further, we analysed the violations with respect to flow seasons. For this, we classified

each month into low (Q < 0.4MAF), intermediate (0.4MAF ≤ Q ≤ MAF), and high (Q > MAF) flow classes, in which MAF stands for mean annual flow. This classification was based on the flow season limits in the EF methods selected for this study (Table 1), and it aims to illustrate the dependency between the amount of discharge and EFE violations. For each GHM, we computed the flow season of each month from median discharge across all GCMs. MAF was computed from the respective year of each month, so that individual months could be classified into different seasons during different years, thus

accommodating for drier and wetter years.

Further, we conducted a seasonal trend analysis on the recent historical EFE violation frequency and severity. For the trend analysis, we computed the frequency and severity of violations according to the definitions above, but instead of all years (1976–2005), we applied five-year moving windows starting from the first window 1976–1980 and ending in the last window

2001–2005. Each of the moving windows was computed over four GHMs and consisted of 240 violation ratios (4 GHMs x 5 years x 12 months). Then, for each sub-basin and separately for frequency and severity, we computed the Kendall rank correlation coefficient and fitted a linear regression model into the moving window series (n = 26). We eliminated statistically non–significant (p > 0.05) trends using the Kendall rank correlation test (Hollander and Wolfe, 1973) and the linear regression slope t–test (Chambers et al., 1990).


Finally, we performed a fuzzy c-means clustering (Bezdek, 1981) for each flow season separately. The four clustering variables constituted violation frequency, violation severity, and the linear trend slopes associated with each. We chose to create six clusters, of which the most likely one was selected for each sub-basin. If no cluster was selected with over 30% likelihood for

a sub-basin, that sub-basin was left unassigned. The output of the cluster analysis is a set of sub-basin clusters, within which intra-cluster similarity and inter-cluster dissimilarity are maximised. Therefore, in sub-basins belonging to one cluster, the EFE violation characteristics and trends are similar.

# 3 Results

## 3.1 Recent historical and pre-industrial EFE violations

Our findings show that (1) EFE violations are widespread around the world, (2) that lower bound violations are more common than upper bound violations, and (3) that the most impacted regions are located mainly in the arid and dry temperate climate zones (Fig. 2a–c). All of the results presented in this section include only violations in a minimum of three-month streaks, which emphasises persistent flow alterations and masks short-term variation (see Sect. 2.3). The EFE is violated in 49.8% of the total 3,860 sub-basins during more than 5.0% of the total 1,440 months between 1976 and 2005 (4 GHMs x 360 months) (Fig. 2a). At this threshold, the violations have more than doubled compared to the pre-industrial period (Fig. A4a). The EFE lower bound is violated in 43.2% of sub-basins during more than 5.0% of all months (Fig. 2b), whereas the respective figure for the EFE upper bound is 9.6% (Fig. 2c). Regional patterns are more visible in the EFE lower bound violations than in the EFE upper bound violations, as sub-basins showing lower bound violations are more commonly grouped together. Notable EFE violation patterns emerge in areas with high anthropogenic pressure, such as the Middle East, India, Eastern Asia, and Central America. As the violation frequency shown in Fig. 2a–c is computed over all 1,440 months, it corresponds to the unweighted ensemble mean of the four GHMs. The most impacted regions also remain comparable for individual GHMs, of which PCR-GLOBWB shows the least frequent and LPJmL shows the most frequent violations, with H08 and WaterGAP2 falling in between these (Fig. S4).

For a comparison between the pre-industrial and recent historical periods, we computed the change in violation frequency between them (Fig. 2d–f; Fig. A4g–i). During the pre-industrial period, the EFE is violated in 24.0% of all sub-basins during more than 5.0% of all months (Fig. A4a). The majority of this consists of EFE lower bound violations, as no sub-basins have more than 5.0% of months violated solely due to upper bound violations (Fig. A4c). Since EFE violations prevail in certain regions also during the pre-industrial period, some of the violations can be assumed to be caused by natural variability. However, the EFE violation frequency has widely increased since the pre-industrial period (Fig. 2e), which indicates remarkable changes in discharge. These changes are highlighted when counting sub-basins with more than 10.0% of all months violated (32.7% recent historical; 9.6% pre-industrial) or 25.0% of all months violated (9.5% recent historical; 0.08% pre-industrial). While the EFE lower bound violation frequency has been considerable – especially in the driest mid-latitudes and Australia during the pre-industrial period (Fig. A4b) – many of these regions have also experienced the largest increases in violation frequency (Fig. 2e). Conversely, parts of the Northern Hemisphere show a slight overall decrease in EFE lower bound violation frequency (Fig. 2e).

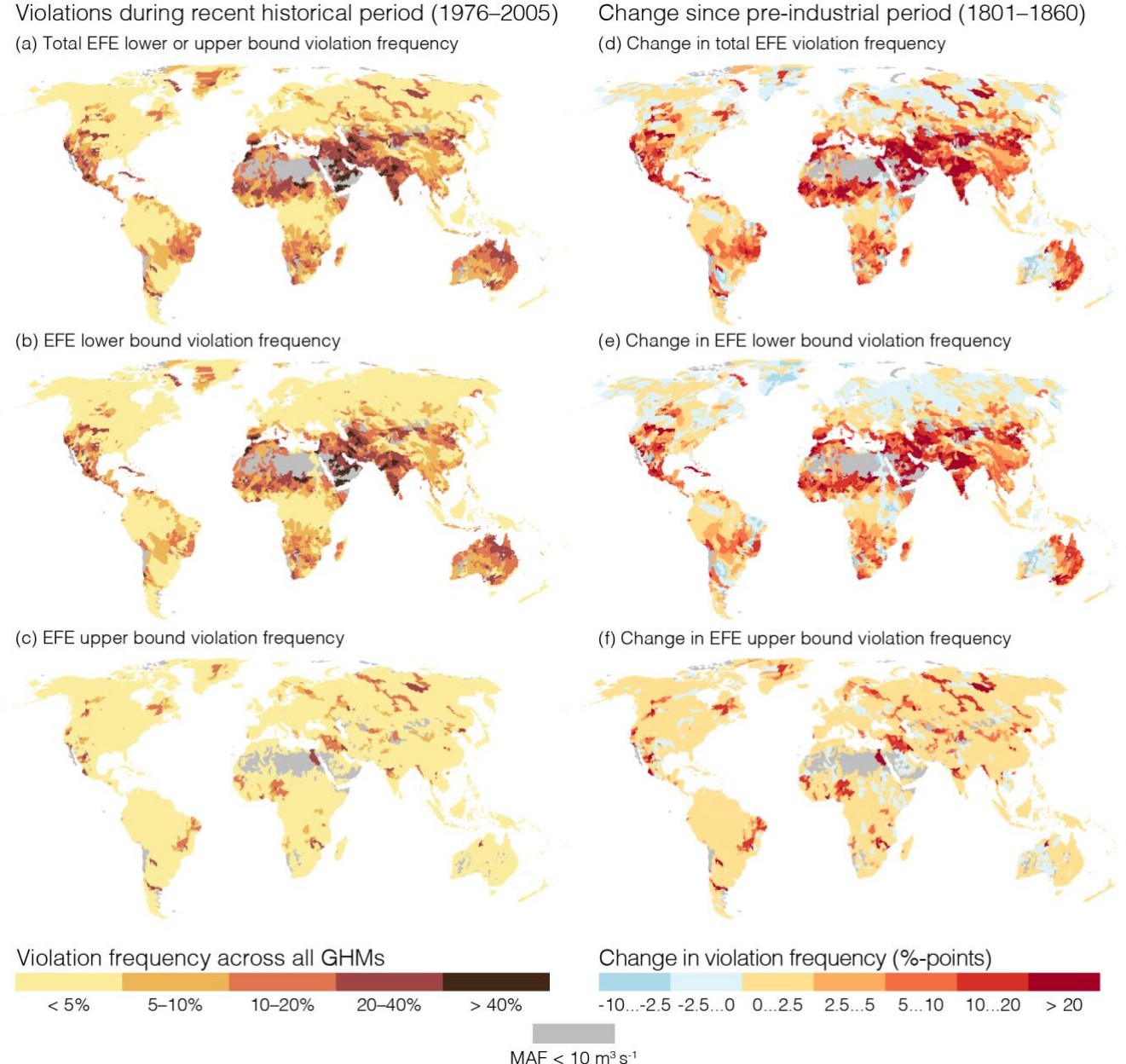

**Figure 2**: Frequency of environmental flow envelope (EFE) violations during the recent historical period (1976–2005); for both upper and lower bounds (a), lower bound only (b), and upper bound only (c), aligned with the change in violation frequency since the pre-industrial period (1801–1860) (d–f). All values are computed across four global hydrological models (GHMs). Sub-basins with mean annual flow (MAF) less than 10 m³ s⁻¹ at the sub-basin outlet are excluded. Case examples on how the recent historical discharge compares to the EFE in different flow regimes are shown in Fig. A3.

### 3.2 Seasonal characterisation of recent historical EFE violations

The low-flow season is clearly the most impacted in terms of EFE lower bound violations, while the violations decrease gradually from low- to intermediate-, and intermediate- to high-flow seasons (Fig. 3a–c). The distinction between flow seasons is stronger for the frequency than the severity of violations. Between 1976 and 2005, the EFE has been violated in 83.5%, 59.0%, and 28.6% of sub-basins during low-, intermediate-, and high-flow seasons for at least one three-month streak (frequency > 0). The medians of EFE lower bound violation severities for low-, intermediate-, and high-flow seasons are -37.1%, -19.0%, and -24.7%, respectively. These values mean that the typical EFE lower bound violation is caused by discharge falling 19–37% below the EFE lower bound. Although violation severity appears to be less dependent on flow season compared to the dependency of violation frequency on flow season, the low-flow season remains the most impacted overall. This is supported by the spatial coverage of sub-basins in the class of the most frequent (> 25%) and the most severe ($Q < 0.5EFE_{lower}$) violations, which reaches across all continents during low-flow season (Fig. 3c) and decreases in prevalence during intermediate- and high-flow seasons (Fig. 3a–b). While the spread in EFE lower bound violation frequency and severity is notable between GHMs (Fig. S6), especially the distinction between flow seasons is clearly visible in all single-GHM results (Fig. S5).

EFE upper bound violations are less dependent than EFE lower bound violations on flow season and exhibit less consistent spatial patterns of frequency and severity (Fig. 3d–f). The shares of sub-basins within which the EFE upper bound is violated for at least one three-month streak between 1976 and 2005 are 15.5%, 24.6%, and 18.9% for low-, intermediate-, and high-flow seasons respectively. The medians of EFE upper bound violation severities during low-, intermediate-, and high-flow seasons are 153%, 121%, and 123%. Although the summarised statistics would suggest typical EFE upper bound violations to be caused by discharge exceeding the EFE upper bound by 21–53%, many of the sub-basins experiencing EFE upper bound violations fall into the high-severity categories, within which discharge exceeds the EFE upper bound at least twofold (Fig. 3d–f). These extremes often cover a small number of sub-basins at a time (Fig. 3e–f; e.g. Tigris-Euphrates river system, northern China, Niger River), whereas larger-scale patterns covering more sub-basins show less frequent and less severe EFE upper bound violations (Fig. 3d–e; e.g. north-eastern Europe, Central Asia). For individual GHMs, the spread in EFE upper bound violation frequency and severity is substantially higher than for EFE lower bound violations (Fig. S8). Most of the EFE upper bound violations originate from other models except PCR-GLOBWB, but the three other models show fair agreement in identifying the sub-basins with major EFE upper bound violations (Fig. S7).

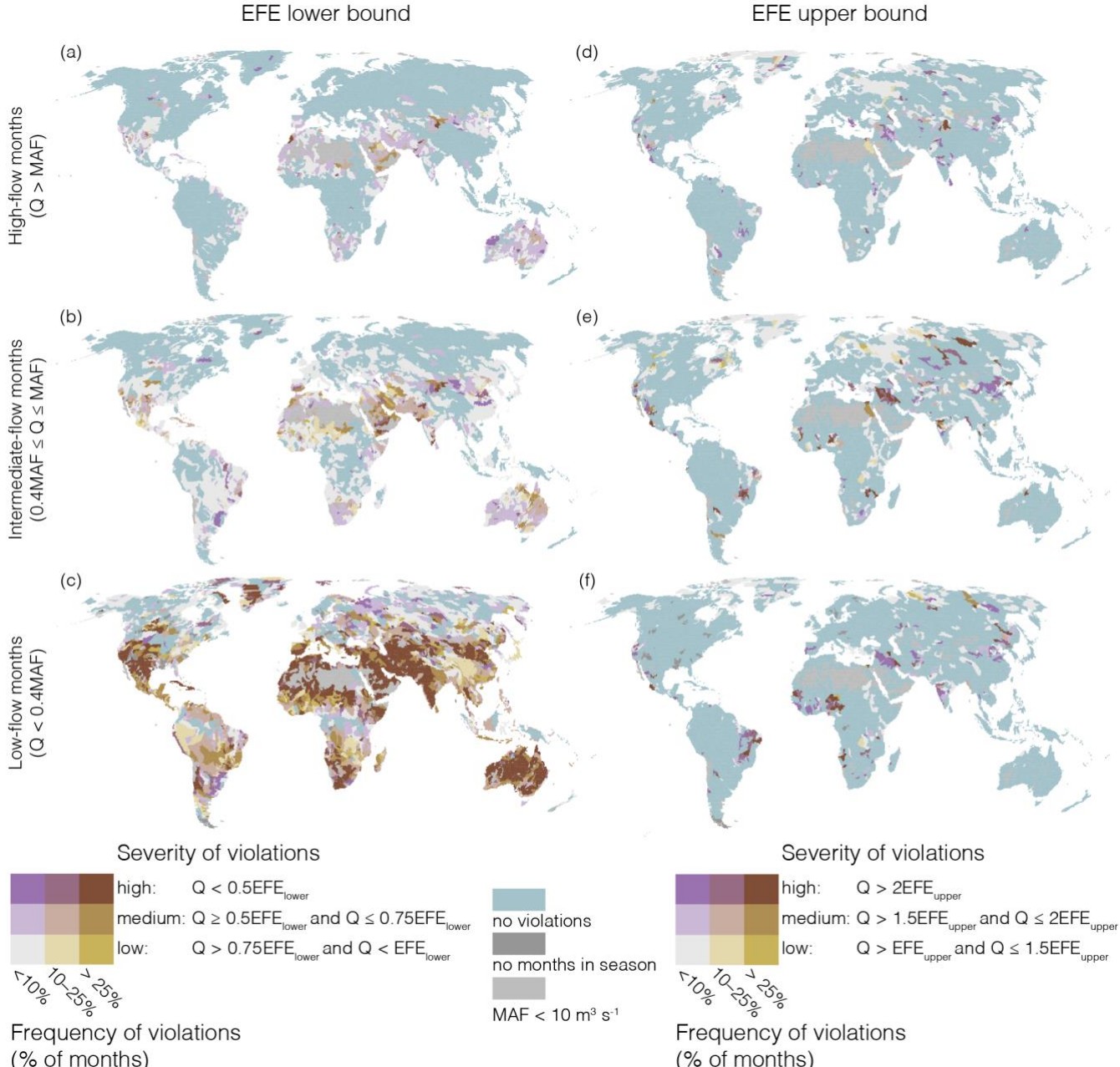

**Figure 3**: Seasonal frequency and severity of environmental flow envelope (EFE) violations of the lower bound (a–c) and the upper bound (d–f). Q stands for monthly discharge and MAF for mean annual flow. For each sub-basin in each season, violation frequency corresponds to the fraction of violated months out of all months in that season, and violation frequency to the mean violation ratio during those violated months. If there are no months between 1976 and 2005 during which discharge would fall below the low-flow season limit, the respective sub-basin is classified as "no months in season".

### 3.3 Trends in recent historical EFE violations

Between 1976 and 2005, the frequency and severity of EFE violations have often increased or decreased together. Although we are unable to analytically determine the main drivers of this, we show that more sub-basins have experienced amplifying rather than attenuating EFE violation trends. For the EFE lower bound violations, a statistically significant violation trend is observed for 15.0–51.9% of all sub-basins depending on flow season (Fig. 4a–c). This violation trend consists of a statistically significant trend in violation frequency, severity, or both. Many of the trends (41.0–64.8% of all detected trends depending on flow season) consist of a frequency and a severity trend in the same direction – i.e. both violation frequency and severity are increasing or decreasing. For the EFE upper bound violations, 10.3–16.6% of all sub-basins show statistically significant violation trends, and most of the trends (68.4–72.1%) consist of changes in the same direction (Fig. 4d–f). Conflicting violation trends are rare; trends consisting of an increase in one variable and a decrease in the other cover 0.5–5.4% of all detected trends across both EFE bounds and all three flow seasons.

The agreement between the direction of EFE violation frequency and severity trends highlights that the violation trends co-develop rather than conflict. This also holds when computing trends for individual GHMs. Though a statistically significant trend is identified by all individual GHMs in relatively few sub-basins, cases in which some GHMs would detect increasing and some decreasing trends are rare (Fig. S11). Since increasing violation frequency combined with increasing violation severity is the single most common trend for both EFE lower and upper bound violations (28.4% and 53.1% of all detected trends across all flow seasons), the general trend of EFE violations has been towards the intensifying direction during the recent historical period.

In most of the world, the trends of EFE lower and upper bound violations are independent, but signs of EFE violation trends shifting from the lower bound to the upper bound can be identified – especially in the Northern Hemisphere and the Pan-Arctic areas. Trends in which the EFE lower bound violation frequency and severity are decreasing prevail, for example, in parts of Russia and northern Canada (Fig. 4c), yet the same regions show increasing trends in EFE upper bound violations (Fig. 4e). Therefore, while the EFE lower bound violations can be alleviated by increasing discharge, the EFE upper bound violations may be amplified at the same time. The net EFE violations may then balance out, or the EFE upper bound violations may in turn dominate and increase the overall violations. For most of the world, however, this shifting of violations is not visible, and trends – as well as the EFE violations overall – concentrate on one boundary of the envelope only.

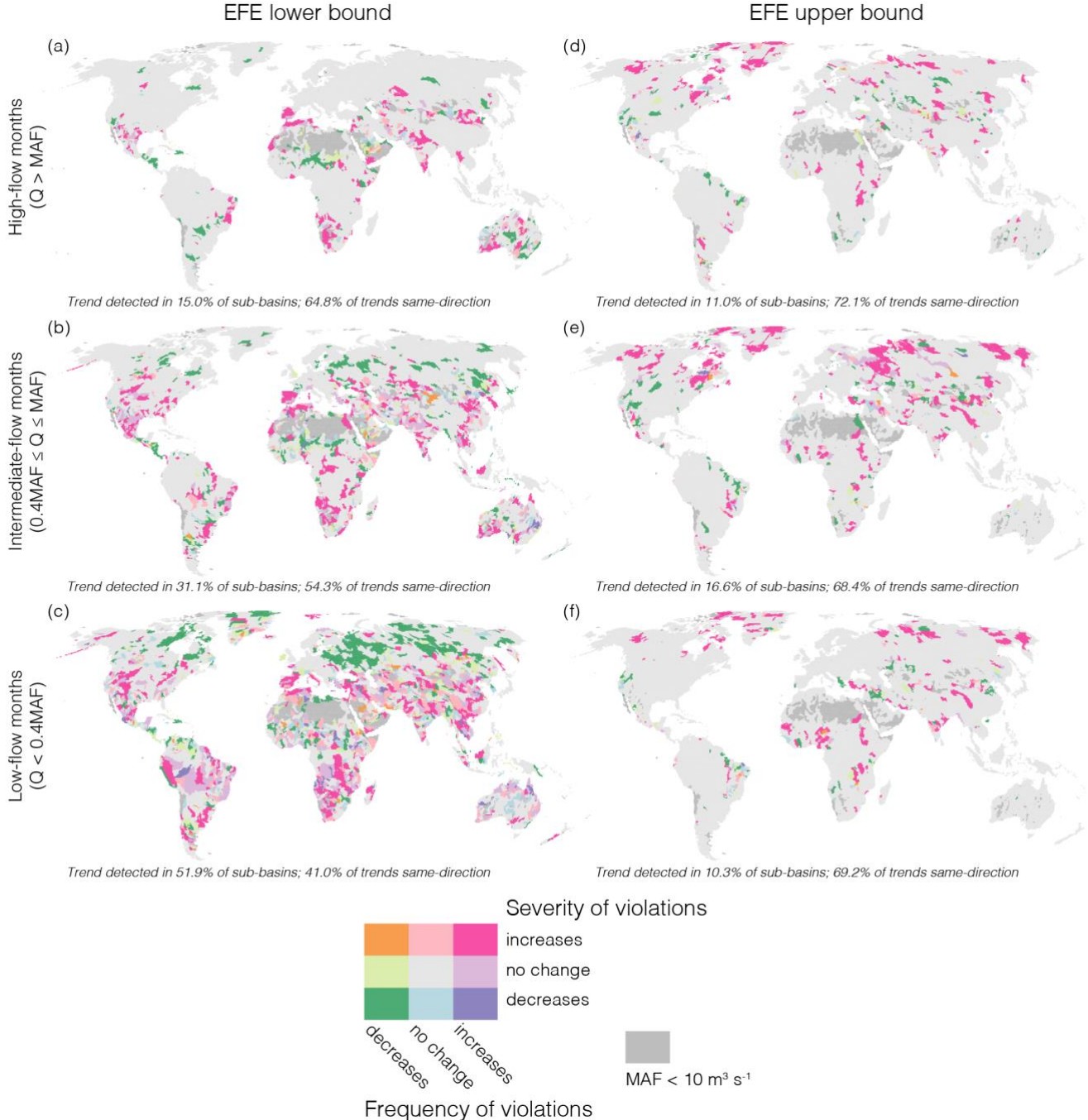

**Figure 4**: Trends of frequency and severity of environmental flow envelope (EFE) violations of the lower bound (a–c) and the upper bound (d–f). Q stands for monthly discharge and MAF for mean annual flow. The trends are computed using five-year moving windows from 1976 to 2005; only statistically significant trends are shown. Same-direction trends are defined as having both frequency and severity trends in the same direction, i.e. both violation frequency and severity are increasing or decreasing. The steepness of the trend slope is not considered here but the trends are classified only by increasing or decreasing direction.

## 3.4 Categorisation of sub-basins by recent historical EFE lower bound violations and trends

The arid mid-latitudes along with parts of tropical South America and subtropical Africa and Asia emerge as the most impacted regions in terms of EFE lower bound violations when the frequency, severity, and trends associated with both are combined in a seasonal cluster analysis. In the relative paucity of sub-basins experiencing EFE upper bound violations, we performed the cluster analysis for the EFE lower bound violations only (for further details, see Sect. 2.3). In Fig. 5, the seasonal clusters are grouped together into cluster groups and named according to the characteristics of EFE lower bound violations within each group. The blue cluster group A encompasses areas with very few violations, while the first EFE violations appear during low-flow season in the turquoise cluster group B. EFE violation frequency and severity increase in the purple and yellow cluster groups C–D compared to A–B. The orange cluster group E consists of sub-basins with highly variable EFE violation characteristics, and the red cluster group F corresponds to the areas with the highest frequency and severity of EFE lower bound violations within each season.

The cluster groups A–C represent sub-basins with minor or stable EFE violations, while EFE violations in sub-basins belonging to cluster groups D–F can be considered the most remarkable throughout all flow seasons. Adding to the nearly non-existent EFE violations in the cluster group A, the cluster group B shows minor violations with decreasing low-flow season trends. Although the cluster group C shows relatively common violations, sub-basins in this group are not experiencing amplifying trends. The regions previously identified as the most impacted (Sect. 3.1–3.2) are mainly covered by cluster groups D–F. Specifically, sub-basins within the yellow cluster group D currently experience moderate violations but show the steepest increasing trends in both violation frequency and severity during low-flow season, and increasing trends during other seasons. If past trends continue, these sub-basins – which include densely populated regions in Asia as well as regions in South America with rich riverine ecosystems – may be under the most serious threat of intensifying EFE violations.

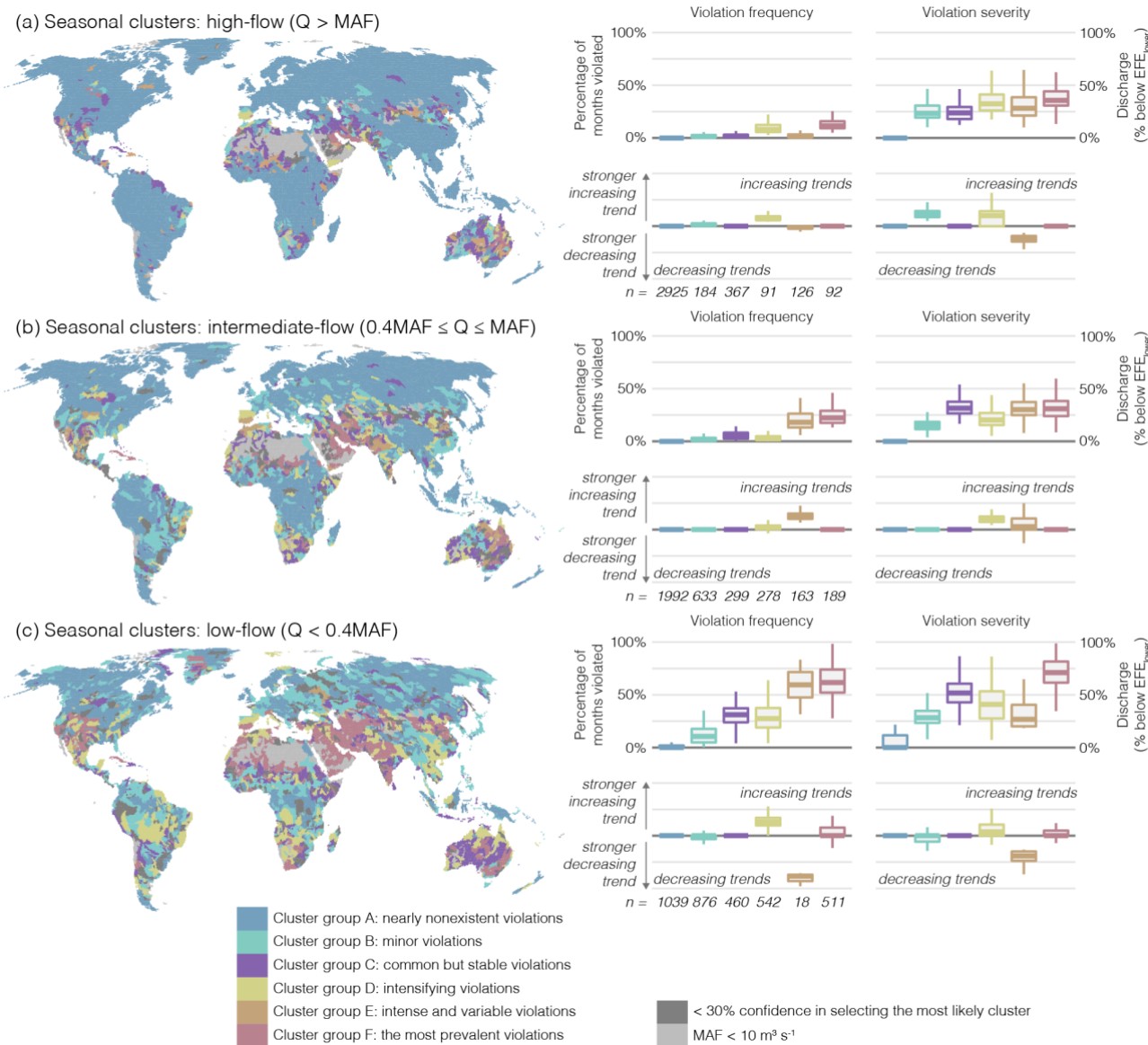

**Figure 5**: Seasonal clustering of sub-basins based on environmental flow envelope (EFE) lower bound violations during high-flow season (a), intermediate-flow season (b), and low-flow season (c). Q stands for monthly discharge and MAF for mean annual flow. The four clustering variables constitute violation frequency, violation severity, and linear trend slopes associated with both variables. In the trend slope box plots, the position of each box is proportional to the linear trend slope; boxes further away from the middle line indicate steeper trend slopes and thereby more rapid change in violation metrics. Sub-basins with mean annual flow (MAF) less than 10 m³ s⁻¹ at the sub-basin outlet were excluded from the clustering.

## 4 Discussion

In this work – which is the first to quantitatively address frequency, severity, and trends of EFE violations combined – we show that the EFE is widely violated across the globe between 1976 and 2005 (Fig. 2a). Given that the change from the pre-

industrial period is substantial (Fig. 2d) and all considered violations last three or more months (Sect. 2.3), the EFE violations represent persistent flow alterations during the recent historical period. The EFE lower bound violation patterns are strongly seasonal, with the low-flow season being the most affected in terms of both frequency and severity of violations (Fig. 3a–c). The EFE upper bound violation patterns are more dispersed and harder to characterise (Fig. 3d–f). Some sub-basins have experienced drastic flow alteration, with discharge either falling to half of the EFE lower bound or increasing to more than double the EFE upper bound (Fig. 3). Further, recent historical trends in EFE violations have been amplifying rather than attenuating, showing increases in both violation frequency and severity (Fig. 4). Combined, our results show that many sub-basins in the most densely populated and ecologically diverse areas – such as East Asia, South Asia, and parts of South America – are already experiencing considerable EFE lower bound violations, and these can be expected to intensify based on past trends (Fig. 5).

## 4.1 Comparison to existing studies

EFE violations in the arid mid-latitudes, India, Eastern Asia, and the west coast of North America compare well to the EF violations estimated by Jägermeyr et al. (2017). These are also in line with the areas requiring the largest reductions in water withdrawals to meet EFRs (Droppers et al., 2020). Our EFE violations are more widespread in large parts of Australia, South America, and Southern Africa (Fig. 2–3) than those reported by Jägermeyr et al. (2017). However, Jägermeyr et al. (2017) determine EFRs based on pristine discharge simulation between 1980 and 2009 and report annual averages, which differs from our seasonal analysis that is based on the pre-industrial period and includes potential climate change impacts. Central Europe and parts of North America show minor EFE violations (Fig. 2–3; Jägermeyr et al., 2017), although rivers in these regions are highly fragmented, regulated, and threatened or even degraded in terms of biodiversity (Grill et al., 2015, 2019; Grizzetti et al., 2017; Vörösmarty et al., 2010). Regarding EF violation magnitude, Jägermeyr et al. (2017) report discharge deficits mainly under 10%, whereas our results show substantially higher violation severities (Fig. 3a–c).

The key benefit of our ensemble approach (see Sect. 2.2) is that the ensemble metrics can be considered globally feasible (Arsenault et al., 2015; Huang et al., 2017). Since using the ensemble counters the hydrological uncertainty always embedded in GHMs (Telteu et al., 2021), our results could be assumed to be more robust compared to studies using single GHMs or EF methods (e.g. Gerten et al., 2020; Hoekstra and Mekonnen, 2011; Pastor et al., 2019; Steffen et al., 2015). Although the ISIMIP 2b data are representative of historical anthropogenic drivers including dams and reservoirs (Frieler et al., 2017), the inclusion and parameterisation of human impacts invokes substantial uncertainty in GHM outputs, particularly in terms of flooding and dam operation (Masaki et al., 2017; Veldkamp et al., 2018). Regarding the Pan-Arctic areas in particular, GHMs have recently been shown to perform relatively poorly (Gädeke et al., 2020), which calls for cautious interpretation of our results in these regions. As the individual GHMs show notable differences in EFE violations (Fig. S4–S8) and the spread in ensemble EFRs is often substantial (Fig. A2; Hogeboom et al., 2020; Jägermeyr et al., 2017; Liu et al., 2021; Pastor et al., 2014), selecting an

array of GHMs and EF methods would be highly desirable in global EF assessments to understand and address the related uncertainties.

## 4.2 Key drivers of recent historical EFE violations

Three key drivers of EFE violations can be identified from existing global research: increasing water use (Graham et al., 2020; Müller Schmied et al., 2016), flow regulation, especially by dam operation (Döll et al., 2009; Schneider et al., 2017), and the indirect impact of climate change on streamflow (Arnell and Gosling, 2013; Asadieh and Krakauer, 2017; Gudmundsson et al., 2021; Moragoda and Cohen, 2020; Thompson et al., 2021; Wanders et al., 2015). Major EFE violations prevail in the densely populated mid-latitudes, in which anthropogenic impacts often dominate long-term streamflow alterations (Müller Schmied et al., 2016). The anthropogenic impacts on flow alteration are also reflected in the projected increase of water stress (use-to-availability ratio) that is driven primarily by increasing water use (Graham et al., 2020). Factors beyond the sub-basin scale, such as water use in upstream sub-basins (Munia et al., 2020) or land use induced changes in atmospheric moisture recycling (Wang-Erlandsson et al., 2018), can further affect the net anthropogenic flow alteration within a sub-basin.

In the subtropical Southern Hemisphere, increasing EFE lower bound violation trends can be expected to co-occur with the projected trends of increasing droughts (Asadieh and Krakauer, 2017; Wanders et al., 2015), as both indicate abnormally low amounts of water in a system. On the other hand, the decreasing EFE lower bound violation trends and the increasing EFE upper bound violation trends in high-latitude Europe and Siberia (Fig. 4b–e) may link to climate change–induced changes in discharge (Arnell and Gosling, 2013; Asadieh and Krakauer, 2017). Though our 30-year recent historical period is relatively short for identifying climate trends, Gudmundsson et al. (2021) report decadal trends already within this period. However, climatic changes are not the sole cause of EFE upper bound violation trends; for example, dam operation can increase discharge especially during low-flow season and therefore result in EFE upper bound violations (Döll et al., 2009; Poff et al., 2017; see also Fig. A3b). This can be seen in EFE upper bound violations located above Boguchany and Krasnoyarsk mega-dams in Siberia (Fig. 2c; Lehner et al., 2011).

## 4.3 Relationship between EFEs and riverine ecosystem integrity

Our method – and EFs in general – assumes that violating or respecting the EFE is associated with degrading or preserving riverine ecosystems. However, this might not hold for simplified hydrological EF methods as they lack metrics of assessing the correlation between presumably adequate hydrological conditions and ecosystem responses (Poff and Zimmerman, 2010; Richter, 2010; Richter et al., 2012; Mohan et al., 2021). Practical discharge allocation based on insufficient EF methods has even been argued to potentially cause further degradation of riverine ecosystems (Arthington et al., 2006; Shenton et al., 2012). This is because the ecosystem response to altered flow regimes varies across spatiotemporal scales and different species due to the impact of altered flow regimes on e.g. sediment transport, stream and riparian bank morphology, and community dynamics of fauna and flora (Biggs et al., 2005; Poff and Zimmerman, 2010; Poff et al., 1997; Rolls et al., 2018). Our results

support the assertion that the ecological representativeness of hydrological EF methods is limited, as they show minor violations in many ecologically impaired areas (Sect. 4.1).

Recently, quantitative water flows have been shown to be less important than water quality and invasive species for assessing rivers' ecological status, determining fish biodiversity, and driving fish habitat loss (Barbarossa et al., 2021; Grizzetti et al., 2017; Su et al., 2021). Though fish make up only a part of a riverine ecosystem, these findings underline that discharge alone cannot provide a comprehensive EF definition but other factors should be considered, as well. Holistic EF methods that include these factors – and also observation of biotic responses – correlate much better with ecosystem states, but require in situ data, ancillary variables, and local expert knowledge (Poff et al., 2017; Tharme, 2003) that are not available at the global scale. For practical use and analyses of sub-basin scale riverine ecosystem integrity, our results should be complemented by local studies using holistic EF methods. However, our globally consistent approach using hydrological EF methods provides a comprehensive global overview on anthropogenic flow alteration.

Our selection of the 95th percentile of pre-industrial discharge as the EFE upper bound is only a first step towards a more informed choice (Sect 2.2). The link between EFE upper bound violations and ecosystems exists, since, for example, floodplain ecosystems in monsoon flood pulse systems require distinct dry and wet periods, and disturbing the dry period by increased discharge may degrade the ecosystems (Hayes et al., 2018; Junk et al., 1989; Schneider et al., 2017). However, case studies that would quantify this link are scarce. Here, we intentionally set the EFE upper bound very high by drawing it from a distribution that contains potentially very high discharge values (Sect. 2.2). Resulting from this, the EFE upper bound is exceeded very rarely during the pre-industrial period (Fig. A4c). Hence, EFE upper bound violations are strong signals of increased upper extreme flows, although it cannot be inferred from this study whether these are detrimental to riverine ecosystems beyond regions with distinct dry and wet periods, such as the monsoon areas. We stress that our proposal is not meant to be established as a global presumptive standard, but rather as a first trial to inspire methodological advances grounded in hydro-ecology.

## 4.4 Limitations and way forward

The EFEs have their specific limitations related to temporal and spatial scales. Temporally, we aggregated daily discharge to monthly discharge, incurring a loss of temporal detail – especially regarding extreme high and low flows. However, we consider this necessary to assess persistent EFE violations, which is also the rationale behind enforcing the three-month consecutive violation streak rule (Sect. 2.3). Moreover, to prevent raising the EFE upper bound extremely high, we excluded outlier discharge prior to determining the EFEs (Sect. 2.2), which may result in excluding not only potential model errors but also extremely rare natural events. Spatially, we consider the sub-basin outlet location as representative for the whole upstream area, which simplifies the sub-basin into one hydrological unit (Sect 2.1). Using the coarse grid scale could potentially provide unstable results due to high GHM variability in headwater and low-discharge streams, which is countered by this aggregation.

However, the aggregation also masks local EFE violations that may vary within the sub-basin itself. This is a notable limitation particularly in the case of temporary rivers, which have recently been shown to comprise a large part of global rivers, and which are highly important for local ecosystems (Messager et al., 2021). In addition, HydroSHEDS level 5 lumps small coastal sub-basins with many estuaries together, which in our analysis results in many small catchments to be represented by the outlet location of the largest lumped catchment. Therefore, our results are the most robust in relatively large sub-basins with a single outlet channel. Applications at the scale of small catchments consisting of few 0.5-degree grid cells should rather resort to high-detail observed data instead of global data simulated in a coarse grid.

In the future, global analysis with more advanced EF methods and more detailed hydrological data that better correlate with riverine ecosystem status could further develop the EFEs. Furthermore, case studies quantifying riverine ecosystem responses to prolonged and increased upper extreme flows would benefit the development of the EFE upper bound, as the mechanism is known (Hayes et al., 2018; Junk et al., 1989; Schneider et al., 2017) but quantitative methods are lacking. While the main drivers of EFE violations are recognised here, a systematic analysis on the couplings between them and EFE violations would provide more insights into our results. Separating natural and anthropogenic flow alterations could prove useful in estimating how major violations – and in which regions – should be deemed as the most serious. This way, the actions to alleviate EFE violations could be best prioritised and targeted to the most affected regions. Our EFE methodology is lightly parameterised and applicable with open global data sets – the availability and quality of which is constantly increasing. While fine-tuning is required for local contexts, the EFEs provide a quick and globally robust way of assessing the extent and degree of flow alteration.

## 5 Conclusion

Direct and indirect anthropogenic flow alterations have drastically changed flow regimes across the world and are likely to threaten riverine ecosystem integrity. In this study, we have conducted a global, in-depth analysis of the flow alterations using environmental flow envelopes (EFEs). The widespread and long-standing flow alterations found can be expected to be amplified in response to projected future increases in human water use, the building of new dams, and climate change. Operationalising our results at the basin scale requires more detailed data, assimilation of cross-scale information, and interdisciplinary knowledge to more fully portray the ecological and hydrological conditions of each unique river. Nevertheless, our results highlight the need to consider environmental flows in both global research and policies on water resources management as major anthropogenic flow alterations prevail across wide areas.

## Appendix A. EFE conceptualisation and assessment

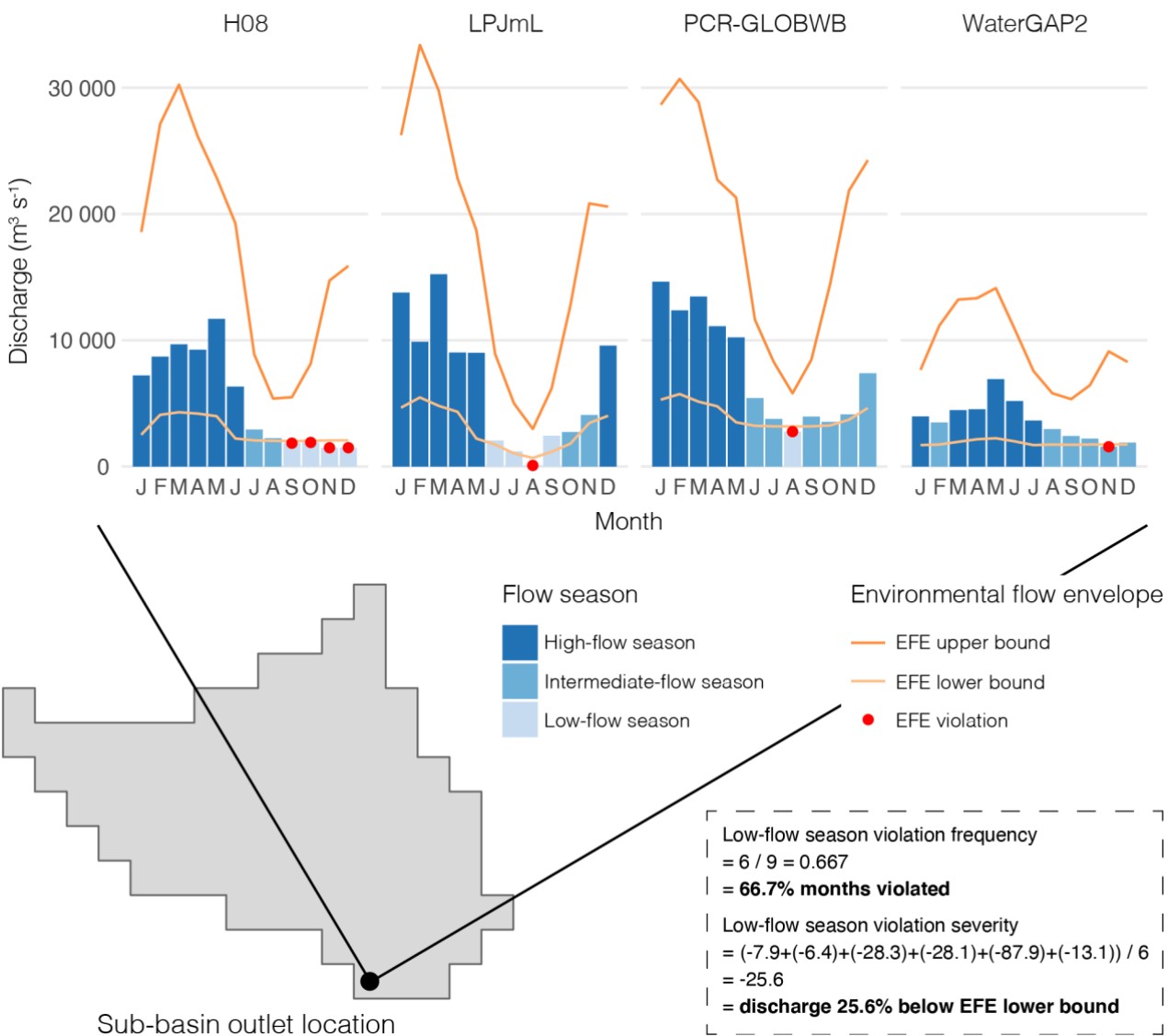

**Figure A1**: Case example on the conceptual definition of the environmental flow envelope (EFE) and the assessment of EFE violations. The example sub-basin is a part of the Rio Paraguay basin: the outlet is located a little upstream from Asunción, Paraguay. For simplicity, we show discharge and assess EFE violations only for the EFE lower bound and one year. In addition, we do not enforce the 3-month violation streak rule (see Sect. 2.3) in this example but count all individual violated months. If the 3-month rule was enforced, violations from the H08 model only would be counted. For each global hydrological model (GHM; H08, LPJmL, PCR-GLOBWB, and WaterGAP2), the discharge is the median of four general circulation models (GCMs). The EFE violation frequency and severity are computed according to definitions in Sect. 2.3.

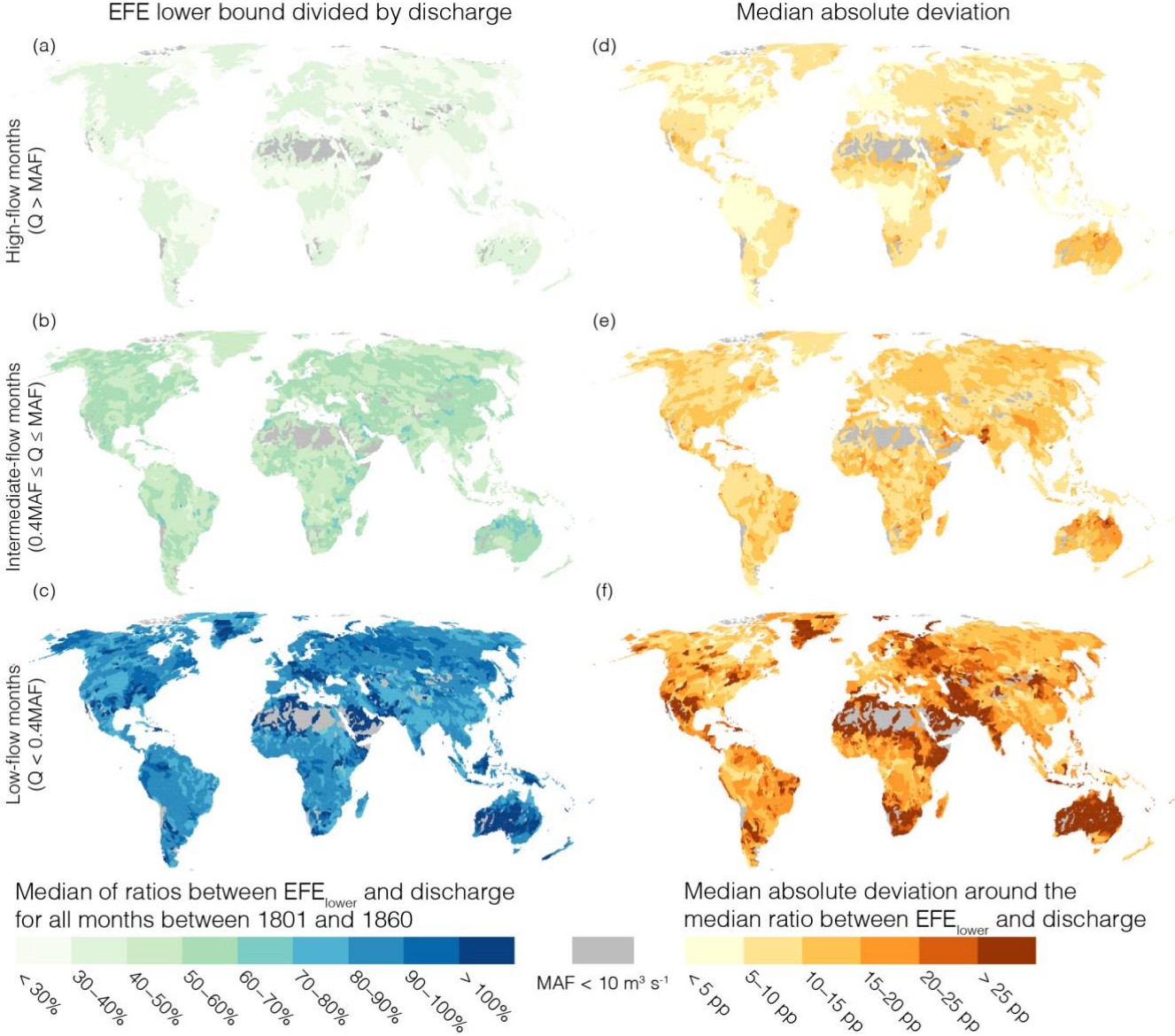

**EFE lower bound divided by discharge**

**Median absolute deviation**

Median of ratios between EFE$_{lower}$ and discharge for all months between 1801 and 1860

< 30%  30–40%  40–50%  50–60%  60–70%  70–80%  80–90%  90–100%  > 100%  MAF < 10 m³ s⁻¹

Median absolute deviation around the median ratio between EFE$_{lower}$ and discharge

< 5 pp  5–10 pp  10–15 pp  15–20 pp  20–25 pp  > 25 pp

560

**Figure A2**: Comparison between the environmental flow envelope (EFE) lower bound and pre-industrial discharge. Q stands for monthly discharge and MAF for mean annual flow. Here, for each global hydrological model (GHM) and month, we took the pre-industrial median discharge over all general circulation models (GCMs) and divided the EFE lower bound with it, yielding a total of 2,880 ratios for each sub-basin (4 GHMs x 60 years x 12 months). Then, for each season and across all GHMs, we took the median of the resulting EFE$_{lower}$ / Q ratios
565 (a–c) and computed the median absolute deviation around this value (d–f). Some EFE lower bound values exceed the median low-flow season discharge due to high variation in pre-industrial discharge affecting the distribution of environmental flow requirements (EFRs), from which the EFE lower bound is drawn (see Sect. 2.2). Moreover, the spread of ratios between EFE lower bound and low-flow season monthly discharge is relatively high, further indicating high variability in low-flow season discharge modelled by GHMs during the pre-industrial period.

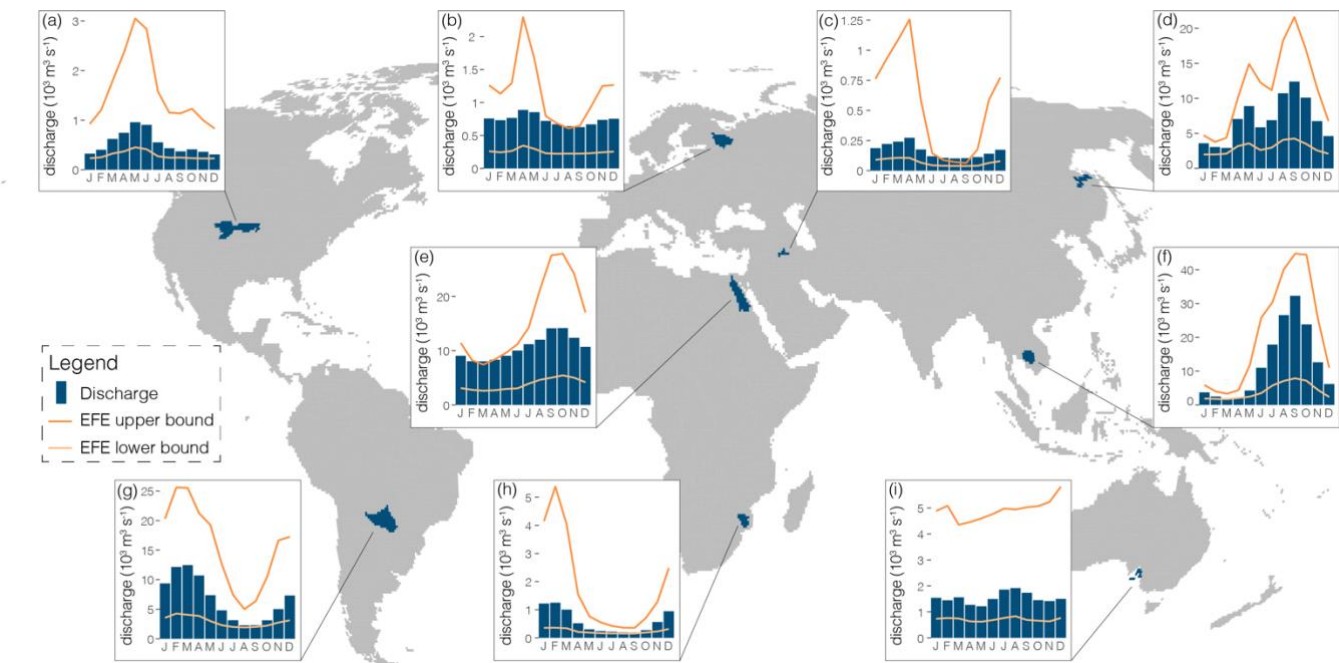

570

**Figure A3**. Case examples of environmental flow envelopes (EFEs) and mean monthly discharge in variable flow regimes. For the sake of illustration, we show both EFE lower and upper bounds as mean values of four global hydrological models (GHMs). Accordingly, the discharge presented here is the mean monthly discharge between 1976 and 2005, computed from four discharge data sets from four GHMs. Further, for each GHM, the discharge is the median of four general circulation models (GCMs) as outlined in Sect. 2.3. The anthropogenic

575 flow alteration is clearly visible in some of these sub-basins: for example, the spring peak flow in Fig. A3b has decreased whereas summer flows have substantially increased compared to pre-industrial EFEs.

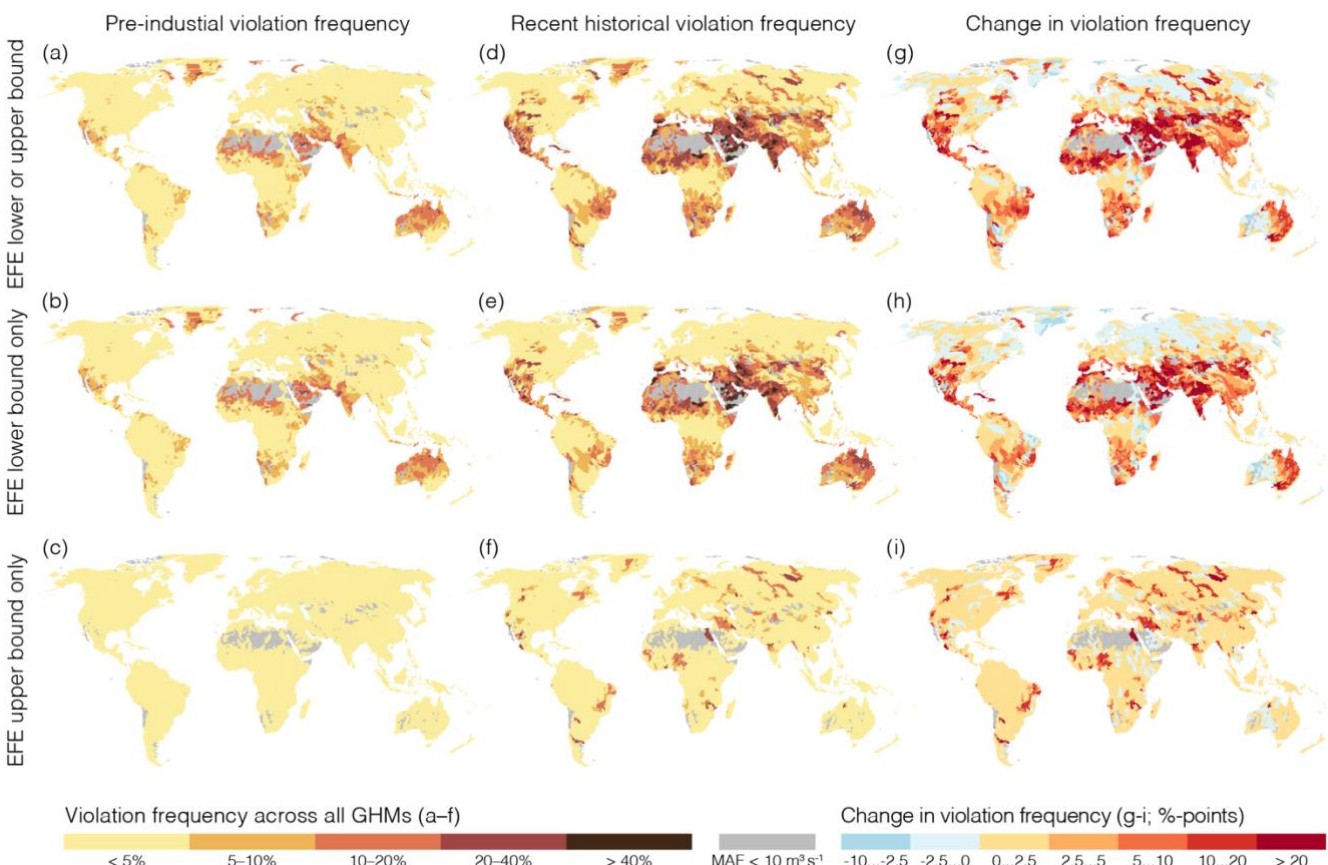

**Figure A4**. Frequency of environmental flow envelope (EFE) violations between 1801 and 1860 (a–c), between 1976 and 2005 (d–f) and the change in violation frequency between these two periods (g–i). The maps are separated by violations of both upper and lower bounds (a, d, g), lower bound only (b, e, h), and upper bound only (c, f, i). All values are computed across four global hydrological models (GHMs). Sub–basins with mean annual flow (MAF) less than 10 m³ s⁻¹ at the sub-basin outlet are excluded.

## Code and data availability

The code and data used in this study are available in https://zenodo.org/record/6552573.

## Author contribution

MK, EA, and VV conceptualised the study with input from MP, LA, TG, CM, LWE, and DG. DG, MF, NH, HMS, and NW performed the ISIMIP simulations, which were coordinated by SNG and HMS. EA processed the raw data, wrote the implementation of the EFE methodology, and conceived the initial analysis with help from VV, MP, LA, and MK. VV revised and performed the final analysis and produced the results and visualisation shown in the study, discussing together with MK, MP, LA, TG, CM, and LWE. VV wrote the manuscript based on EA's work with contributions from all authors.

## Competing interests

The authors declare that they have no conflict of interest.

## Acknowledgements

We would like to thank the ISIMIP team and all participating modelling teams for making the outputs available, and Amy Fallon for her diligent proofreading of the manucsript.

VV was funded by the Aalto University School of Engineering Doctoral Programme. EA and LA were funded by Maa- ja vesitekniikan tuki ry. MK received funding from Academy of Finland funded project WATVUL (grant no. 317320) and European Research Council (ERC) under the European Union's Horizon 2020 research and innovation programme (grant agreement No. 819202). MP was funded by the European Research Council (ERC) under the European Union's Horizon 2020 research and innovation programme (grant agreement No. 819202), and by the Erling-Persson Family Foundation. LWE acknowledges financial support from the European Research Council through the "Earth Resilience in the Anthropocene" project (no. ERC-2016-ADG 743080).

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
