# Peer review of "Widespread and increasing violations of environmental flow envelopes"

_Hydrology and Earth System Sciences, 2021_

## Referee Comment (RC2)

**HESS Review of:**
**Virkki et al. 2021 Environmental flow envelopes: quantifying global, ecosystem–threatening streamflow alterations**

**General:**
Title implies ecosystem-threatening alterations are quantified - is this really the case given the lack of ecosystem assessment within the analysis?

English grammar will need a full check and improvement, as some of the points made are hard to follow as written. Shorter, sharper sentences will help.
e.g. Abstract li 39 - The use of less and few.
Introduction li 50 - "rivers upkeep two major…"; li 84 - data are plural

The use of language is strong in some cases, perhaps intentionally?  - e.g. threatening, violations - this does influence the tone of the MS.

**Abstract:**
li 27 - "We present a novel method to determine EFs by Environmental Flow Envelopes (EFE), which is an envelope of variability bounded by discharge limits within which riverine ecosystems are not seriously compromised."
The authors need to take care not to overstate this point.  Presenting the approach as novel is highly problematic in that a sustainability boundaries/envelope approach to eflows was proposed over a decade ago, e.g. by Richter (2009) and then Richter et al. (2012) based on concepts presented in Postel and Richter's book Rivers for Life (2003).  The application of the approach globally, at finer basin scale, and the attention to frequency, severity, and trends combined, are arguably where there is novelty in this paper.
In addition, the claim that ecosystems would not be seriously compromised will not be supported by evidence and should be tempered.

Some recent/key references need consideration, incl:
Richter, B. D., M. M. Davis, C. Apse, and C. Konrad. 2012. A presumptive standard for environmental flow protection. River Research and Applications 28:1312-1321.
Poff, N. L., C. M. Brown, T. E. Grantham, J. H. Matthews, M. A. Palmer, C. M. Spence, R. L. Wilby, M. Haasnoot, G. F. Mendoza, K. C. Dominique, and A. Baeza. 2016. Sustainable water management under future uncertainty with eco-engineering decision scaling. Nature Climate Change 6:25-34.

li 30 "considering also the methodological uncertainties related with global EF studies".  The logical connection between this phrase and the rest of the sentence is not clear.

li 31. "EFE introduces an upper bound of discharge, identifying areas where streamflow has increased substantially."  Such sentences are poorly expressed and need rewriting for technical substance, as well as sharper understanding; increased above what? Presumably natural levels?

li 41 - Because of the way in which you define a violation, you need to compare the result of 5% with the pre-alteration state.  Also, the "discharge" cannot "violate" - you need to revise grammar for correctness in these sorts of sentences, perhaps by saying EFE violations.

li 44 - The potential attribution to climate change (cf. other human stressors impacting upper bounds of the flow regime) - supposition, or do you show this analytically using data?

**Introduction:**
This section needs at least moderate revision.

li 55 - are 5 references needed for this point?  The authors need to strike a better balance between under- and over-referencing in this section.

li 65 - I suggest you also emphasise other aspects of regime variability, including intra-annual and interannual variability.

li 66 - the importance of flow extends well beyond physical habitat.  See e.g. Bunn and Arthington (2002).  In fact, beyond the very general references, this paper lacks attention to ecological dimensions, despite the title suggesting otherwise.
Bunn, S. E., and A. H. Arthington. 2002. Basic principles and ecological consequences of altered flow regimes for aquatic biodiversity. Environmental Management 30:492-507.

li 76 -  You present the 2018 definition which focuses on a regime that meets a set of objectives, and then regress to the now out of date concept of a minimum discharge in the following "refer to the minimum discharge required to sustain healthy and functional riverine ecosystems (Pastor et al., 2014)."  Try to draw out the regime-based argument rather.

li 77 - "Hence, the EFR corresponds to a boundary not to be transgressed."  This is overly simplistic in the way it is stated.

li 77 - "Beyond simple EFRs, more nuanced quantification of anthropogenic impacts on discharge based on a multitude of different metrics include e.g. the Indicators of Hydrological Alteration (IHA; Richter et al., 1997, 1996)."  This is a poor explanation.

li 80 - The EF assessments are not implemented in legislation as such.  Legislation that provides for EF is one of the factors that enables EF to be implemented.

li 84 - Thereby, global studies accommodating EFs…rather use… considering it as a…well‑being"  This is not a well structured sentence for clarity or grammar.  There are many such examples in the text.

li 109-110.  There is potential in the approach to undertake such analyses of trends.  However, it is overstating the case that EFEs are new as a concept.

li 111 "envelope of safe discharge variability that…"  That it is safe or even precautionary is really an assumption and that needs to be clear.  You are using a composite of hydrological methods that are all limited in their ecological relevance and representation.  You do not undertake any actual ecological assessment either, so you need to take real care not to overstate the case.

li 113 - I suggest you reference a paper that explains hydrology methods, past and present, ecologically relevant flow metrics etc. and is more up to date than the IHA refs you provide
e.g. Poff et al. (2017)

Poff, N.L., Tharme, R.E. and Arthington, A.H.  2017.  Evolution of environmental flows assessment science, principles, and methodologies.  Chapter 11.  Pp: 203-236.  In: Horne, A.C., Webb, J.A., Stewardson, M.J., Richter, B. and M. Acreman, (Eds).  *Water for the environment: from policy and science to implementation and management*.  Academic Press.

**Methods and Data:**
All assumptions and caveats e.g. violations, need to be well explained.

The technical argument for "violations" is weak, particularly in relation to the comparisons of near-natural (largely unmodified by humans), present-day and future flow regimes. A more rigorous explanation is needed.  One of the main challenges with the approach and the terminology is with the violation of Q90.
With an eflow of Q90, that flow will be naturally 'violated' 10% of the time. Given the authors' choice of a low flow eflow that is the median value of hydrological methods ranging from Q90 to the mean monthly value of dry months, the so-called 'violations' will occur even more frequently.  It is thus unclear as to what proportion of the 'violations' described are simply a result of the hydrological approach used.  This information needs to be provided, as any violations resulting from human impacts on the flow regime should be in excess of the natural ones.

li 150 - You need to provide referencing for HydroBASINS when it is first mentioned, especially as its use is critical for the sub‑basin divisions applied. You only provide the reference at li 150. The introductory paragraphs before section 2.1 could focus more on the general steps.  Suggest you reword for clarity, so that there is no confusion about what is meant by "into consistently sized" and the hydrological nested approach can be well understood.

Reasons for the selection of models need to be give, for the hydrological, climate change and the five hydrological eflow methods.

Greater explanation of the advantages and limitations of the input datasets from ISIMIP2b is needed.

Similarly, the reasons for the selection of 1801–1860 as the reference flow period need to be expanded on.  What is meant by "quasi-natural"? How was this ascertained, or was it assumed based on the pre-industrial era? Likewise, for the present-day/recent data period.  What about flow alteration post 2005?  For some regions/basins very recent trends are significant - what are the implications of not including this most recent part of the time series?
There are also several good references addressing criteria for hydrological record selection that are surprisingly not mentioned in this section on methods.

li 153-155.   You actually exclude a fairly significant number of basins - did you test what the implications might be?  At the least you need to objectively acknowledge this fact and its implications, rather than word it in such a way as to downplay it.

li 163 - The EF methods.  The strengths and limitations of these methods are not provided, but need to be clear to the reader.  Table 1 just lays out the methods.  There are a number of more recent hydrologic methods that are less simplistic and/or include a greater diversity of ecologically meaningful flow metrics.  You have not used any of those methods in the composite approach.  The

reasons need to be given, especially given the assumptions made in the paper around ecological threat/safety.

The paper would benefit from some discussion of why more sophisticated methods including regional approaches (ELOHA based) were not possible to apply at this time.

Li 180-183. This section of text needs careful attention. "As the EFE lower bound, we selected the median of the EFR distribution…potentially consist of unrealistically low or high EFR estimates, caused by either highly deviant discharge provided by certain GCMs or distinctively different representation of ecosystem water needs in the EFR method". These two points need to be broken down and carefully argued. They are based on assumptions without any apparent ecological rationale. This needs to be clear, e.g. is the choice of median as the lower bound arbitrary? It designates the low-high flow boundary sometimes hydrologically, but that is not the same as a boundary designated for eflows purposes. Also, the last point on "distinctively different representation of ecosystem water needs" is hard to follow - what are you intending to say here?

Li 207. "we excluded time periods during which the EFE is violated for less than three consecutive months"- by definition you would then exclude floods. Can that be correct?

li 213. " excluded sub‑basins with extremely low flow from our analysis" This is a limitation that warrants more explanation or returning to in the discussion section. You also need to clarify the extent to which temporary rivers might be excluded as a result and to discuss the implications (see new Nature paper by Messager et al. 2021 on the global importance of such systems) Messager, M.L., Lehner, B., Cockburn, C. et al. Global prevalence of non-perennial rivers and streams. Nature 594, 391–397 (2021). https://doi.org/10.1038/s41586-021-03565-5

li 214. You excluded basins if the mean flow was less than 10 cumecs – that is rather high as so many regions have the bulk of their rivers smaller than this. Interestingly the resolution of your maps does not show these smaller exclusions. By way of illustration – the Mgeni River in South Africa supports 6 million inhabitants in two cities Pietermaritzburg and Durban. The river just above the estuary in its natural condition only exceeded 10cumecs for 38 days a year, so this entire river should be excluded from your map – yet it does not show as excluded. This is concerning and should be discussed.

li 225 - "classified each month of record into low ($Q < 0.4MAF$), intermediate ($0.4MAF \leq Q \leq MAF$), and high ($Q > MAF$) flow classes". On what basis? What evidence is here from the literature to support this categorisation or was it somewhat arbitrary?

li 234. Statistical procedures require supporting references. e.g. for Kendall rank correlation coefficient.

li 236 - "Finally, we combined the EFE violation frequency and severity throughout the recent past time series with the linear violation trend slopes and performed a fuzzy c‑means clustering (Bezdek, 1981) to each flow season separately." This sentence needs breaking down into clear points. Reasons need to be given for choices made e.g. for fuzzy c-means clustering. This is important. e.g., as Fig 5 relies heavily on the clustering method adopted.

**Results:**

li 239 - "EFE violations are widespread around the world, concentrating on lower bound violations in the arid and dry temperate climate zones"  The pre-alteration context needs to be given, for comparison and for these results to have meaning.  Also, there are limitations of the approach for rivers with very low to no flows, and this needs covering in the results and discussion.
Again, there are issues around grammar and the expression of information that detract from the sense of this sentence (violations that concentrate on lower bound violations).  While I understand what is intended, such sentences need to be corrected.

li 245.  "Therefore, the EFE is rather violated by insufficient than excessive discharge, and regional patterns are more clearly visible in EFE lower bound 245 violations whereas EFE upper bound violations are more dispersed into individual sub‑basins." I think the results need breaking down further to make this argument.  How much of this pattern is a result of: using hydrology based approaches that tend to focus more on the low flow regime (Q90) or monthly time step, and/or less on event-based influences such as floods; or a regional effect attributable to naturally more arid conditions rather than a function of the operation of infrastructure?

How sensitive are the results to the somewhat arbitrary setting of the upper and lower bounds, and to natural flow regime characteristics?

li 273 - Niger River - for a named river, the R should be upper case. Tigris-E Basin?  Northern China - the N might need to be lower case unless it is a specific region.

li 259-276.  The results are interesting and some of them could feature more substantively in the discussion.

li 287 - "Respectively…"Such sentences are poorly structured and compound, and need to be rewritten for clarity.

li 300 - "increasing discharge alleviating EFE lower bound violations may turn out to be amplifying for EFE upper bound violations in some regions and downplay the positive indications of decreasing EFE lower bound violation trends."  Such sentences cannot be understood as written.

**Discussion:**
The discussion needs greater depth in content, including in relation to ecological limitations, and the implications of the results.

Draw in some points on more arid river systems (where the approach seems particularly weak) and on temporary rivers.

li 346 - Suggest you differentiate between ecosystem change and threat - e.g. in Europe many rivers are already ecologically degraded, not simply at threat of degradation.

li 348 - "Since these areas show relatively little EFE violations, it can be inferred that even though the quantitative discharge would be within the EFE, the anthropogenic flow alteration can still be major."  Should read "relatively few/minor".  This is a key point that needs more elaboration - otherwise, it suggests that the approach taken/methods used are insufficiently ecologically representative or attuned to reality.

li 359 - "can largely be attributed to anthropogenic impact…" But one might argue that you simply see potential congruency rather than having demonstrated an attribution/causal link through the data?

li 362 - What is meant by "remotely teleconnected" here?

Li 366. This is suggesting that drought is the main cause of increasing violations – not increased abstraction? That is strange as demands in these regions are high.

li 384 - "flaws" - suggest reword as "limitations" and ref. Poff et al. 2017 here. If they are flawed, then this puts their use into question.

li 384 - "global fish biodiversity has shown that several other factors, such as water quality and the presence of invasive species, may be more important in maintaining riverine ecosystems than quantitative flow (Su et al., 2021)." Yes, but the context is important to get right here and in general. Moreover, fish are one group of species, not directly equivalent to an ecosystem or its functions. This section on ecology is weak and needs a very careful read-through to address assumptions/oversimplifications.

li 386 - "well-being" is typically used for people, and integrity or health for ecosystems.

li 394-404. This is a mix of natural flow paradigm, novel and hybrid systems, designer flows, shifting baselines etc. and needs to be sorted out into a more cogent argument.

li 396 - "its globally equal absoluteness". This is not English as written.

li 408 - Section 4.3 needs a serious edit. Please use simpler sentences that make clear points, to help with the logic of your argument.

li 409 - You need to provide a reference or better introduce "ensemble thinking" in this context - here or under the methods.

li 427. Terms such as "excessive flows" need to be expressed using more scientifically appropriate wording.

li 447 - "quantification of the riverine ecosystem responses to prolonged and excessive flows through case studies would benefit the development of the EFE upper bound." Suggest include some supporting references here for a couple of e-flow case studies that address this aspect.

Li 455 – it would be useful to comment on the application of this approach at the local scale compared to the global scale, and the limitations associated with that. In line 466 you mention "operationalising our results at the basin scale", but just what this means could be clarified.

li 466. Can you be more specific on ways to make the approach more meaningful operationally i.e. from an implementation standpoint? There are obvious limitations, given the characteristics of the EF methods used. So the appropriateness and the limits of such simple global scale, hydrology-based methods must be kept in mind in these concluding remarks.

**Figures and tables:**
The paper follows a nice, clear and logical layout and the figures are generally good.

Fig 3. It is difficult with these colours to differentiate the low violations cases from the less than 10% frequency case. Also, is it well explained in the text as to what is meant by the shorthand "no months in season"?

Fig A1 - There is too much small text in the figure.

**Recommendation**
Moderate revision.

---

## Author Response (AR1)

**Referee comment 1**

**RC 1.1. General comments**

The authors quantify the environmental flow violation for most worldwide sub-basins using environmental flow envelopes (EFE). EFE are based on previously established minimum environmental flow requirements and a newly established maximum environmental flow requirement. EFE quantification was based on pre-industrial simulations of several global hydrological models (GHM), under various global circulation models (GCM), and applied to historical simulations for each GHM. Results inform about the frequency and severity of EFE violation, and could thus indicate the state of riverine ecosystem security.

Environmental flow is a concept often used in hydrological modelling in order to simulate sustainable water use scenarios that do not interfere with riverine ecosystem security. Moreover, they can be a useful tool to determine sustainable dam and water management policies. The authors aim to reduce the uncertainty in the current (historical) violation of these requirements, and to expand the violation assessment to the worlds sub-basins, by using a GHM-GCM model ensemble. However, I have some concerns regarding the EFE quantification methods, the attribution of EFE violation, and some general comments

We thank the Referee for the comments, which are helpful and will certainly help to revise the paper stronger. We are happy to see that the overall story of the paper has been conveyed well. Please find point-by-point responses to the comments below.

**RC 1.2. EFE quantification methods**

The authors aim to advance environmental flow assessments in four ways: (1) improve the environmental flow concept to include an upper bound (line 95, line 114, line 186, line 458), (2) reduce the uncertainty in the actual quantification of environmental flows, in particular the large GHM uncertainties (line 87, line 92, line 112, line 409), (3) use the pre-industrial river streamflow as the environmental flow basis (line 116, line 160, line 460), and (4) present the quantification of environmental flows in terms of frequency, severity and trends (line 98, line 115, line 458).

**RC 1.2.1.** While I understand the adverse effects of upper EFE violation, the selection of the upper bound seems arbitrary. What is the reason for the 95th percentile?

**AC 1.2.1.** There is evidence that naturally dry periods - which could be compromised by long-standing increased flows -  are required for certain ecosystems (see also lines 92-96). However, methods to quantify this are lacking, and, to our knowledge, no globally applicable method exists yet. In the absence of quantitative case studies, we use the 95th percentile as a

simple indicator of substantially increased flows. We assume that under natural conditions, this would rarely be exceeded in periods lasting three or more consecutive months. We found this true according to our newly performed analysis in response to RC 2.5.1., which means that the EFE upper bound violations originate from the time between the pre-industrial and recent past periods. However, it is possible that adverse ecosystem effects begin already below and above this bound. We will expand the justification and discussion on the upper bound in the revised manuscript, and clarify that the upper bound is provisionally set in the absence of methods and case studies of the ecological relationship. Our aim is to provide a first analysis and inspire future methodological advances more based in hydroecology, instead of establishing a presumptive global standard.

**RC 1.2.2.** I feel this study does not properly address the GHM uncertainty. The ensemble strategy is only applied to the EFR methods and GCMs, not the GHMs. EFE violations are still determined for each model separately, without averaging, which can lead to additional (compounding) uncertainties. Moreover, if the aim of the authors is to provide accurate quantifications of EFE for each river, why did the study not use observational data or select the "best" performing GHM model for each basin?

**AC 1.2.2.** We chose to determine the EFEs and EFE violations for each model separately so that we only compare the recent past discharge from one GHM to EFE defined by the same GHM. If we compared recent past discharge and EFEs from different GHMs, the results could be more affected by GHM structural differences than changes in discharge.

When we combine the GHM-wise results into the results shown in the manuscript, we do create a GHM ensemble as well. As noted in line 218, we compute the frequency of EFE violations from a total of 1,440 violation ratios from four GHMs (360 ratios each). The same result would be achieved if we computed the frequency of violations for each GHM separately, and took the unweighted mean of this. Therefore, the violation frequency shown in Fig. 2 corresponds to the ensemble mean of the four GHMs. The seasonal violation frequency and severity are not directly equal to the ensemble mean of GHMs as the count of violated months varies between GHMs and flow seasons. We will more explicitly elaborate on these metrics in the revised manuscript, and provide more detail on single-GHM results.

We opted not to select the "best" performing GHM for each basin against observational data because the "best" model varies largely based on the sub-basin characteristics and also based on the metric which is used in assessing the models (Zaherpour et al., 2018). We also consider that a comprehensive evaluation of discharge in over 4,000 basins globally is out of the scope of this article, and it would detract readers from the key contributions of the article. However, in the absence of discharge validation, we consider best to estimate EFEs for all models separately and supplement the ensemble results with the GHM-wise results (see AC 1.3.2.). Any further use of the EFEs, especially at the sub-basin scale, warrants performance evaluation with observational data. The application at the local scales is also discussed in AC 2.6.16.

**RC 1.2.3.** What is the rationale behind using pre-industrial river streamflow? Should the simulated historical pristine (no human influence) river streamflow not be better to estimate EFE (as is done by other modelling studies)? It is briefly discussed that historical pristine river streamflow may already be violating EFE (line 399). Should this be the case?

**AC 1.2.3.** Indeed, other studies, such as Jägermeyr et al. (2017), use simulated historical pristine river streamflow to estimate natural state environmental flows. However, adopting pre-industrial river streamflow as the baseline accounts for both change in climate and change in human water use. If we adopted historical pristine river streamflow, we could account only for changes in human water use and the most recent climatic changes, omitting natural or human-induced variability in climate since the pre-industrial period. It is correct that the EFE can be violated during pristine streamflow conditions, which we have now analysed in response to RC 2.5.1. Alongside AC 2.4.5., we will expand our justification of selecting the pre-industrial period as the reference.

**RC 1.2.4.** I am assuming monthly pre-industrial EFE quantifications are combined into a multi-year monthly averages. However, the EFE are applied to historical monthly streamflow values. This could result in EFE violation during dry years (even if EFE were applied during the pre-industrial period). Is this correct?

**AC 1.2.4.** By definition, all of the selected EF methods can be violated by dry years, notwithstanding the cause of dryness being natural variability or human action. Detailed in AC 2.5.1., we have now analysed the pre-industrial EFE violation levels to estimate how much of the violations could originate from natural variability.

**RC 1.3. Attribution of EFE violation**

The authors describe the EFE violation in terms of frequency, severity and trends. To that end they use the gross total of all 1440 historical GHM-year-month combinations (line 232). Subsequently the key drivers are shortly explained in the discussion (line 353).

**RC 1.3.1.** I assume the authors use multi-year monthly averaged pre-industrial simulations to quantify the EFE and subsequently apply these values to the historical period (see my comments above). Can the authors show the impact of this choice on your results? It would be useful to separate the effects of climatic changes (human induced or otherwise) from direct human alterations (e.g. abstractions and dam construction). Moreover, it would be useful to see the temporal variability in EFE violation, especially during dry years.

**AC 1.3.1.** Currently, ISIMIP 2b data does not allow for separation between climatic changes and direct human alterations as simulations of historical pristine streamflow without human impact are unavailable. However, we qualitatively compared our results with literature on the impacts of climate change in Section 4.1., which could indicate areas where the climate impact is greater

than direct human impact. Analytically quantifying and separating the impacts of these two drivers is one of the main ways forward from this study, which we mention in lines 448-450.

Regarding the temporal variability, we are unsure what the Referee precisely aims for commenting here, but we interpret it to consider interannual variation in violations during drier and wetter years. Intra-annually, the low flow season is more prone to violations as shown in Fig. 3. and hence, it could be inferred that overall drier years show more violation. The violations during overall drier years may be caused by natural variability, as violations are visible also during the pre-industrial, natural baseline period (see AC 2.5.1.). However, most of the areas in which the baseline violations are the most prominent (e.g. the Middle East and India) have experienced a substantial increase in violation frequency since the pre-industrial period. Therefore, while a certain baseline of natural violations caused by naturally dry years exists, the change between our time periods has been remarkable.

We opted not to analyse the recent past violations on an annual basis because our main aim was to determine the change that has occurred since the beginning of major anthropogenic flow alteration and human-driven climate change. Further, we characterise the long-term recent past state of violations by focusing on the full 30-year recent past period. Analytically explaining the drivers of violations, including natural and non-natural variability, are left out of the scope of this article and outlined as a way forward. In the revised manuscript, we will expand our discussion on the natural variability of violations during dry years along with addressing the new analysis results introduced in AC 2.5.1. Further, we will emphasise that future research is needed to separate violations caused by natural variability and non-natural variability.

**RC 1.3.2.** Results are given a as the gross total of all GHM-year-month combination. However, the results do not show the variability between the GHMs. Could it be that a majority of the EFE violations originate from a single model?

**AC 1.3.2.** The GHM-wise results, as well as associated dispersion metrics, are shown in the Supplementary material. Of the GHMs, PCR-GLOBWB shows the least violations whereas LPJmL shows the most, H08 and WaterGAP2 falling in between these (Fig. S4). The spread of violation frequency and severity is relatively large, especially in drier regions (Fig. S6, Fig. S8). As there exists dispersion in the violations, the GHM-wise detected trends are not unanimous (Fig. S9-S10). However, cases in which some GHMs would detect decreasing and some GHMs would detect increasing trends are rare (Fig. S11). The regions highlighted in the Results section (e.g. the Middle East, India, Eastern Asia, and Central America) show agreement in violations - as well as in statistically significant trends - in all single-GHM results (Fig. S4, Fig. S11). In the revised manuscript, we will add a brief recap on the differences between GHMs in each Results subsection. Reflecting on the GHM differences further highlights that case-specific auxiliary data is needed if the EFEs were to be applied at the local scales, and adopting a GHM ensemble is highly recommended at the global scale.

**RC 1.3.3.** The authors state that their findings show that EFE violations are widespread around the world (line 239). However, to me the results shown seem quite modest (about half the

basins violate EFE for 5% of the time, not even a month per year). Even more so as dry years could contribute to this number. How do these results change for 10% or 25% of all months?

**AC 1.3.3.** We consider that the violations are widespread around the world because we excluded any violations in streaks shorter than three consecutive months (lines 208-210). This masks short-term changes in flows and highlights long-term changes. When increasing the threshold, we find that the EFE is violated in 32.7% of the total 3,860 sub-basins during more than 10% of the total 1,440 months of record, and 9.5% of the sub-basins during more than 25% of all months of record. In the revised manuscript, we will clarify our argument on why we think the violations are widespread, and incorporate the results with higher thresholds.

**RC 1.4. Specific comments**

**RC 1.4.1.** Line 110 and 658: "The significantly advanced global environmental flow assessments with the novel methodology of environmental flow envelopes" is overstated. The concept of EFE is heavily based on previous lower limit environmental flow requirements, while the newly introduced upper limit is unsubstantiated (in this study). Rather the application of the EFE methodology is new.

**AC 1.4.1.** In accordance with AC 2.2.1., we will reformulate the sentence to more precisely indicate the novelties and already existing concepts of this work. It is correct that environmental flow requirements have a long history, and also the concept of defining an envelope has been introduced before (see RC 2.2.1.). However, the global extent and the depth of analysis undertaken in this study are novel, as also pointed out by the reviewer. Furthermore, our extensive uncertainty consideration is more robust than in previous global EF studies (lines 412-413).

**RC 1.4.2.** Line 177 and 178: Why are high and low flows omitted? Should they not be part of the natural flow regime (wet and dry periods)?

**AC 1.4.2.** Here, we did not aim for removing high and low flows, but we rather aimed for removing extreme outliers. Following this aim, we chose to exclude monthly discharge further than three standard deviations away from mean monthly discharge. Assuming normal distribution, the three-sigma rule retains 99.7% of all simulated discharge values. Hence, 0.15% of the discharge distribution is omitted from both the dry and the wet tail. Discharge falling in the extreme tails corresponds either to true outliers or rare natural discharge events corresponding to a return period of 670 years. We consider that removing true outliers, which could largely affect the computation of e.g. mean discharge, is justified also considering the cost of excluding the extremely rare natural events. In the revised manuscript, we will expand the justification to make our point clearer.

**RC 1.4.3.** Line 241: Do these results still take into account the three consecutive month exceedance threshold (line 210)?

**AC 1.4.3.** All results are shown using the three consecutive month exceedance threshold. In the revised manuscript, we will recap this in the beginning of the Results section.

**RC 1.4.4.** Line 284: Is it implied this trend is due to changes in climate? If so, a thirty year period to detect climate trends is rather short.

**AC 1.4.4.** With this particular sentence, we aimed not to implicate anything related to the cause of trends, and we will revise the sentence not to imply so. We do agree that solely climate-induced trends might not be best detected using a 30-year period, although the study by Gudmundsson et al. (2021) shows that decadal trends are visible already during 1971-2010. This, combined with the future projected changes in discharge are the main arguments why we interpret some of the violation trends to be affected by climate change. However, we can't rule out other factors affecting the violations, which we emphasise in the revised version of this section and Discussion.

**RC 1.4.5.** Line 296: It is interesting to see that the trend in upper bound EFE violation mostly increases in the northern regions, while the lower bound EFE violation mostly decreases in these regions. Do the authors know if this is a climatic or direct human alteration effect?

**AC 1.4.5.** As noted in AC 1.3.1., ISIMIP 2b data does not separate the climatic and direct human alteration effects, so we are unable to analyse this systematically and leave more advanced analysis to future studies. However, our brief review on past changes and future projections in discharge suggest that the northern regions are likely to experience increasing discharge (e.g. Asadieh and Krakauer, 2017). In addition, the most northern regions are sparsely populated and support little agriculture, which indicate little human water use and negligible future water stress as Graham et al. (2020) suggest. On the other hand, some of the northern region sub-basins showing EFE upper bound violations co-occur with large dams, such as Boguchany and Krasnoyarsk dams with 58,200 and 73,300 million $m^3$ storage capacity, respectively (Lehner et al., 2011). Hence, the human impact on EFE violations can't fully be ruled out in sparsely populated regions either, and further quantitative analysis would be required to more explicitly address the drivers. We will qualitatively discuss these drivers of EFE upper bound violations in the revised manuscript.

**RC 1.4.6.** Figure 5: This figure is a little confusing to me. Firstly there is no y axis for the trend slope graph. Secondly there are clusters that have prevalent violations during all seasons, but subsequently do not display violations during for example the high flow months? Lastly, more information could be given about the clustering and why some areas could not be assigned to a cluster.

**AC 1.4.6.** The precise units of the violation frequency and severity trend slopes are left out because all variables are normalised to zero-one interval before clustering. Therefore, the relevant information regarding trend slopes consists only of the direction and relative magnitude of values. Currently, the y axis is labelled with the "increasing" and "decreasing" qualities to

illustrate the direction of trends. The relative position of the boxes further shows how strong the trends are compared to each other. In the revised figure and its caption, we will clarify this.

In addition, we will rename Cluster F to better correspond to the visualisation in which there are only very few sub-basins in this cluster during high flow season. We will also emphasise that the clustering is performed for each season separately.

The reason why some sub-basins could not be assigned to a cluster stems from the clustering method, which gives out probabilities of each sub-basin belonging to each cluster. For a sub-basin, if the maximum probability (the probability of belonging to the most likely cluster) is low, we leave that sub-basin unassigned. Considering also RC 2.4.14., we will include more information on the clustering method to the revised manuscript.

**RC 1.4.7.** line 329: The first section of the discussion could be incorporated in the conclusions.

**AC 1.4.7.** The first section of the Discussion recaps the main findings of the Results section. In the Conclusion section, we outline this once more but with more general expression and no cross-references. We would be hesitant to move the first section of Discussion to Conclusions as it would disable reading the Conclusions section as a standalone text.

**RC 1.4.8.** Line 430: As the aggregated monthly sub-basin assessment restricts the conclusions that can be made, why did the authors not do a daily gridded assessment? The data is available.

**AC 1.4.8.** We conducted a monthly assessment to illustrate and emphasise the longer-standing impacts on river discharge. This is also reflected in enforcing the three-month rule, which omits short-term flow alterations related to extreme events. Furthermore, using daily data at the coarse grid scale could potentially provide unstable results due to high GHM variability in headwater and low-discharge streams. This could, in the worst case, lead to misinterpretations of short-term, high-detail changes based on highly uncertain data and methods. When we simplify the sub-basins to be represented by the outlet cell only (see AC 2.4.10.), we mask the greatest upstream variability and increase the robustness of our sub-basin scale results. However, this causes a loss of detail. In the revised manuscript, we will also highlight the benefits of our selection of spatial and temporal resolution.

**RC 1.4.9.** Line 329 and 461: Where is it shown that the flow alterations reported here are recurrent or long-standing?

**AC 1.4.9.** The argument that flow alterations reported here are recurrent or long-standing stems from the enforcement of the 3-month rule. For justification behind this claim, see AC 1.3.3. We will further emphasise the effect of the three-month rule in the revised manuscript.

**RC 1.5. Technical corrections**

**RC 1.5.1.** Line 51: Should "because of their importance" be removed?

**AC 1.5.1.** The aim of this sentence was to point out that although rivers are known to be important and worth preserving due to their ecosystem regulation capacity, they are also heavily exploited due to the human benefits that can be gained from them. Hence, balancing between these two aims is difficult. We will reformat the sentence in the revised manuscript to make our point clearer.

**Referee comment 2**

**RC 2.1. General**

**RC 2.1.1.** Title implies ecosystem-threatening alterations are quantified - is this really the case given the lack of ecosystem assessment within the analysis?

**AC 2.1.1.** We thank the Referees for the overarching comments on the lack of ecosystem assessment and the potential ways of revising the manuscript. After carefully responding to Referee comments from all three Referees, we consider best to drop the word "ecosystem" from the title and give *"Widespread and increasing violations of environmental flow envelopes globally"* as the new title. We agree that the way in which we treat environmental flows as a proxy for riverine ecosystem integrity is simplistic. Hinting that we would be conducting an ecosystem assessment - while we haven't done any quantitative analysis on it here - may prove misleading to some readers. The new title is better aligned with the key findings and implications of the study.

**RC 2.1.2.** English grammar will need a full check and improvement, as some of the points made are hard to follow as written. Shorter, sharper sentences will help. e.g. Abstract li 39 - The use of less and few. Introduction li 50 - "rivers upkeep two major…"; li 84 - data are plural

**AC 2.1.2.** After editing the manuscript according to the Referee comments, we will thoroughly check and proofread the text to improve its readability and grammar.

**RC 2.1.3.** The use of language is strong in some cases, perhaps intentionally? - e.g. threatening, violations - this does influence the tone of the MS.

**AC 2.1.3.** For these particular words, we intentionally choose to use this language to highlight both the uncertainty and the gravity of the situation. Previous work on this topic has also used such language (e.g. Vörösmarty et al., 2010). As discussed in other comments (e.g. AC 2.4.1. and AC 2.4.8.) and in Section 4.2, it is highly uncertain how well transgressing an EF limit corresponds to ecosystem degradation. Therefore, we choose to use "threaten" as it includes the possibility that EFE violations are not detrimental but the situation is not natural either. Furthermore, the variations of "violate" capture both the lower and upper EFE bounds under one term. As the violations reported in this work are already serious (see AC 1.3.3.), we consider that using this wording is appropriate.

**RC 2.2. Abstract**

**RC 2.2.1.** li 27 - "We present a novel method to determine EFs by Environmental Flow Envelopes (EFE), which is an envelope of variability bounded by discharge limits within which riverine ecosystems are not seriously compromised."

The authors need to take care not to overstate this point. Presenting the approach as novel is highly problematic in that a sustainability boundaries/envelope approach to eflows was proposed over a decade ago, e.g. by Richter (2009) and then Richter et al. (2012) based on concepts presented in Postel and Richter's book Rivers for Life (2003). The application of the approach globally, at finer basin scale, and the attention to frequency, severity, and trends combined, are arguably where there is novelty in this paper.

In addition, the claim that ecosystems would not be seriously compromised will not be supported by evidence and should be tempered.

Some recent/key references need consideration, incl:
Richter, B. D., M. M. Davis, C. Apse, and C. Konrad. 2012. A presumptive standard for environmental flow protection. River Research and Applications 28:1312-1321.
Poff, N. L., C. M. Brown, T. E. Grantham, J. H. Matthews, M. A. Palmer, C. M. Spence, R. L. Wilby, M. Haasnoot, G. F. Mendoza, K. C. Dominique, and A. Baeza. 2016. Sustainable water management under future uncertainty with eco-engineering decision scaling. Nature Climate Change 6:25-34.

**AC 2.2.1.** We agree and will reformat the sentence to more carefully consider which parts of this work are indeed novel and which have been proposed previously in a conceptual manner, as well as incorporate the suggested references. We do agree with the Referees that the conceptual basis of an envelope has been presented previously, but our rigorous quantification of this at the global scale is the main novelty of this study. See also AC 1.4.1.

**RC 2.2.2.** li 30 "considering also the methodological uncertainties related with global EF studies". The logical connection between this phrase and the rest of the sentence is not clear.

**AC 2.2.2.** We will split this sentence into two to make it clearer.

**RC 2.2.3.** li 31. "EFE introduces an upper bound of discharge, identifying areas where streamflow has increased substantially." Such sentences are poorly expressed and need rewriting for technical substance, as well as sharper understanding; increased above what? Presumably natural levels?

**AC 2.2.3.** We aimed to state that the streamflow has increased substantially above pre-industrial levels, which we assume as natural. We will add this to the sentence and reformat it for a better expression.

**RC 2.2.4.** li 41 - Because of the way in which you define a violation, you need to compare the result of 5% with the pre-alteration state. Also, the "discharge" cannot "violate" - you need to revise grammar for correctness in these sorts of sentences, perhaps by saying EFE violations.

**AC 2.2.4.** Regarding the comparison between current and pre-alteration state, we've addressed the comment in AC 2.4.1. and provided the quantitative results in AC 2.5.1. In the revised manuscript, we will incorporate these results also in the Abstract. Further, we will make sure that discharge is not the actor in sentences in which EFEs are violated.

**RC 2.2.5.** li 44 - The potential attribution to climate change (cf. other human stressors impacting upper bounds of the flow regime) - supposition, or do you show this analytically using data?

**AC 2.2.5.** This is an interpretation of the results especially regarding the Northern Hemisphere, where water use can be considered mainly low but EFE violations prevail. We have identified this by comparing our results to literature on climate change impacts on streamflow and addressed it in Section 4.1. and AC 1.4.5. In the revised abstract, we will add a mention that this potential attribution is literature-based.

**RC 2.3. Introduction**

This section needs at least moderate revision.

**RC 2.3.1.** li 55 - are 5 references needed for this point? The authors need to strike a better balance between under- and over-referencing in this section.

**AC 2.3.1.** Here, we have referenced the articles we use in identifying the different drivers of EFE violations later in Section 4.3. First, to establish that streamflow is indeed changing according to observed data, we referenced Gudmundsson et al. (2021). Then, we also show that seasonality is clearly visible in future runoff projections by referencing Arnell and Gosling (2013), which supports our way of separating results to flow seasons. Then, we selected three studies showing that in general, the northern hemisphere is wetting and the southern hemisphere is drying in terms of projected future streamflow. As future projections are uncertain and under continuous development, we chose to incorporate more than one reference to strengthen this point and convey agreement.

**RC 2.3.2.** li 65 - I suggest you also emphasise other aspects of regime variability, including intra-annual and interannual variability.

**AC 2.3.2.** The five factors of natural flow regime mentioned in this sentence are referenced in Poff et al. (1997). We will add a sentence and a reference (Richter et al., 2006) noting that the intra-annual and interannual variability should be considered as parts of the natural flow regime.

**RC 2.3.3.** li 66 - the importance of flow extends well beyond physical habitat. See e.g. Bunn and Arthington (2002). In fact, beyond the very general references, this paper lacks attention to ecological dimensions, despite the title suggesting otherwise.

Bunn, S. E., and A. H. Arthington. 2002. Basic principles and ecological consequences of altered flow regimes for aquatic biodiversity. Environmental Management 30:492-507.

**AC 2.3.3.** We will expand this sentence to cover three other aspects beyond physical habitat listed in Bunn and Arthington (2002), to which we already refer in this sentence. Furthermore, we will complement this sentence with an additional reference (Rolls et al., 2018) on the complex and multifaceted relationship between river flows and ecosystems.

**RC 2.3.4.** li 76 - You present the 2018 definition which focuses on a regime that meets a set of objectives, and then regress to the now out of date concept of a minimum discharge in the following "refer to the minimum discharge required to sustain healthy and functional riverine ecosystems (Pastor et al., 2014)." Try to draw out the regime-based argument rather.

**AC 2.3.4.** The aim of this paragraph is to introduce the more comprehensive ways of defining EFs. However, as we're conducting the study at the global scale with simulated discharge data, we are unfortunately unable to use nothing but the "minimum discharge" definition in the study. To more logically introduce the EF science and its global applications, we will move the "minimum discharge" argument to the next paragraph.

In addition, we will revise the remainder of this paragraph to capture a wider range of advanced EF methodologies, including the general requirement of a collaborative and iterative process in defining EFs (Richter et al., 2006; Poff et al., 2017), as well as ELOHA (Poff et al., 2010), DRIFT (King et al., 2003), and PROBFLO (O'Brien et al., 2018) as examples of holistic EF methods.

**RC 2.3.5.** li 77 - "Hence, the EFR corresponds to a boundary not to be transgressed." This is overly simplistic in the way it is stated.

**AC 2.3.5.** Combined with revision outlined in AC 2.3.4., this sentence will be removed or moved to the following paragraph.

**RC 2.3.6.** li 77 - "Beyond simple EFRs, more nuanced quantification of anthropogenic impacts on discharge based on a multitude of different metrics include e.g. the Indicators of Hydrological Alteration (IHA; Richter et al., 1997, 1996)." This is a poor explanation.

**AC 2.3.6.** We will revise or remove this sentence in accordance with AC 2.3.4.

**RC 2.3.7.** li 80 - The EF assessments are not implemented in legislation as such. Legislation that provides for EF is one of the factors that enables EF to be implemented.

**AC 2.3.7.** We will revise this sentence to convey that legislation is a prerequisite for EFs but EFs are not legislation in themselves.

**RC 2.3.8.** li 84 - Thereby, global studies accommodating EFs…rather use… considering it as a…well–being" This is not a well structured sentence for clarity or grammar. There are many such examples in the text.

**AC 2.3.8.** We will clarify this sentence and potentially divide it into two or more sentences.

**RC 2.3.9.** li 109-110. There is potential in the approach to undertake such analyses of trends. However, it is overstating the case that EFEs are new as a concept.

**AC 2.3.9.** As noted in AC 1.4.1. and AC 2.2.1., we will reformat the novelty statement to not imply that EFEs are new as a concept but the in-depth global scale application is the main novelty, as also pointed out by the third Referee.

**RC 2.3.10.** li 111 "envelope of safe discharge variability that…" That it is safe or even precautionary is really an assumption and that needs to be clear. You are using a composite of hydrological methods that are all limited in their ecological relevance and representation. You do not undertake any actual ecological assessment either, so you need to take real care not to overstate the case.

**AC 2.3.10.** Here, it is indeed an assumption that the "safe" state is related to the minimum discharge thresholds defined by the selected EF methods. We have expanded the justification and use of these methods, considering also the data uncertainties, in AC 2.4.3, AC 2.4.7., and AC 2.4.8. In the revised manuscript, we will make our assumption of "safe" clearer already in the Introduction, and explicitly mention the ecological limitations of our approach.

**RC 2.3.11.** li 113 - I suggest you reference a paper that explains hydrology methods, past and present, ecologically relevant flow metrics etc. and is more up to date than the IHA refs you provide e.g. Poff et al. (2017)

Poff, N.L., Tharme, R.E. and Arthington, A.H. 2017.  Evolution of environmental flows assessment science, principles, and methodologies. Chapter 11. Pp: 203-236. In: Horne, A.C., Webb, J.A., Stewardson, M.J., Richter, B. and M. Acreman, (Eds). Water for the environment: from policy and science to implementation and management. Academic Press.

**AC 2.3.11.** After we have revised the third and fourth paragraphs of the introduction according to AC 2.3.4. and incorporated the suggested reference there, we hope that it is clearer at this point what are the methods of our study and their limitations.

**RC 2.4. Methods and Data**

All assumptions and caveats e.g. violations, need to be well explained.

**RC 2.4.1.** The technical argument for "violations" is weak, particularly in relation to the comparisons of near-natural (largely unmodified by humans), present-day and future flow regimes. A more rigorous explanation is needed. One of the main challenges with the approach and the terminology is with the violation of Q90. With an eflow of Q90, that flow will be naturally 'violated' 10% of the time. Given the authors' choice of a low flow eflow that is the median value of hydrological methods ranging from Q90 to the mean monthly value of dry months, the

so-called 'violations' will occur even more frequently. It is thus unclear as to what proportion of the 'violations' described are simply a result of the hydrological approach used. This information needs to be provided, as any violations resulting from human impacts on the flow regime should be in excess of the natural ones.

**AC 2.4.1.** It is true that the EFE will be violated even during natural conditions (see AC 1.2.4.). However, as noted in AC 1.3.3., we apply the three-month consecutive violation streak rule to exclude one- and two-month violations and show results considering only violation streaks of three or more months. This will mask out violations caused by short-term events and some of those caused by natural variability, resulting in the inclusion of longer violation streaks only. It is correct that even then, some violation streaks may be caused by natural variability (see AC 1.2.4.). Hence, we performed a new analysis in which we computed the violation frequency during the pre-industrial period and compared that to the recent past violations. We have detailed the results of this in AC 2.5.1.

**RC 2.4.2.** li 150 - You need to provide referencing for HydroBASINS when it is first mentioned, especially as its use is critical for the sub–basin divisions applied. You only provide the reference at li 150. The introductory paragraphs before section 2.1 could focus more on the general steps. Suggest you reword for clarity, so that there is no confusion about what is meant by "into consistently sized" and the hydrological nested approach can be well understood.

**AC 2.4.2.** We will provide the reference for HydroBASINS already in the introductory paragraph before Section 2.1 where it is mentioned for the first time. Further, we will revise the introductory paragraph in the beginning of Methods to be written in less technical language and clarify the description of HydroSHEDS data in Section 2.1.

**RC 2.4.3.** Reasons for the selection of models need to be give, for the hydrological, climate change and the five hydrological eflow methods.

**AC 2.4.3.** We selected these models (GHMs) because they are extensively validated against observed data (e.g. Gädeke et al., 2020; Zaherpour et al., 2018). On the other hand, we chose to incorporate more than one GHM as their performance varies substantially (ibid.). In ISIMIP 2b, discharge data is simulated for all four GHMs using four GCMs, which leads us to the natural choice of selecting all available GCMs for each GHM when the spread between GCMs is large (Hattermann et al., 2018).

The EF methods were selected following Pastor et al. (2014) who also validated them against *in situ* EF estimates, finding them adequate. Although more advanced, holistically based EF methods could be available locally, the ones we selected are general enough to be applied at the global scale using global data, but different enough to provide variability to construct an EFR ensemble. The key argument for selecting both the GHMs and the hydrological EF methods is the applicability at the global scale. Finer hydrological data and more advanced EF methods would require substantial resources, which are unavailable for a global study.

In the revised manuscript, we will add a more careful argument on the selection of models and methods under Section 2.1 and 2.2 and provide further details regarding the ensemble approach along AC 2.4.8.

**RC 2.4.4.** Greater explanation of the advantages and limitations of the input datasets from ISIMIP2b is needed.

**AC 2.4.4.** The main advantage of ISIMIP 2b data sets is that the modelled outputs follow a protocol (Frieler et al., 2017). Although the GHMs differ structurally (see e.g. Telteu et al., 2021), the ISIMIP 2b simulation scenario design, including both human and climate factors, is directed by the protocol. Therefore, we can trust that the discharge data is of consistent quality, albeit uncertain. Another advantage is that it provides comparable discharge records since 1801, allowing us to use the pre-industrial baseline to define the EFE.

The main limitation of the ISIMIP 2b data is that the data is of coarse spatial resolution as even state-of-the-art GHMs partaking in ISIMIP 2b operate using the 0.5-degree grid. Furthermore, the uncertainties in such large-scale models are inevitable (Telteu et al., 2021), which is why we consider it necessary to incorporate more than one model as we note in lines 140-142 and AC 1.3.2. In the revised manuscript, we will address the advantages and limitations of ISIMIP 2b data in Methods and Discussion sections.

**RC 2.4.5.** Similarly, the reasons for the selection of 1801–1860 as the reference flow period need to be expanded on. What is meant by "quasi-natural"? How was this ascertained, or was it assumed based on the pre-industrial era? Likewise, for the present-day/recent data period. What about flow alteration post 2005? For some regions/basins very recent trends are significant - what are the implications of not including this most recent part of the time series? There are also several good references addressing criteria for hydrological record selection that are surprisingly not mentioned in this section on methods.

**AC 2.4.5.** There are multiple reasons to justify the selection of the time period 1801-1860 as the reference flow period. First, no significant human modification (large dams, extensive irrigation withdrawals, etc) of rivers had occurred at the time, which enables the assumption of our baseline resembling near-natural conditions. Using the term quasi-natural refers to this assumption. The ISIMIP 2b protocol states that pre-industrial simulations are conducted using fixed land use and socio-economic conditions of the year 1860 (Frieler et al., 2017). These conditions are not fully natural, but very near to it compared to the recent past. Second, we chose to reach far back in time to incorporate the effect of climate change, using a similar assumption that no large human-induced climate change had undergone before or during the reference period (see also AC 1.2.3.). Third, limiting the end date of the reference flow period, the ISIMIP 2b protocol states that pre-industrial simulations end in 1860.

The end of the recent past time series is again restricted by the ISIMIP 2b protocol. The protocol states that simulations that incorporate historical reconstructions of drivers end in 2005. After 2005, the options for ISIMIP 2b data are either fixed-year 2005 drivers or RCP scenarios (Frieler

et al., 2017). As we did not aim for projecting the future, and we considered fixed-year 2005 drivers to inadequately portray the most recent changes, we chose to limit our analysis to end in 2005. For example, historical irrigation extent data ends in 2005 (Siebert et al., 2015) and thus, although there have been a lot of changes since then, no reliable estimates yet exist for all the drivers beyond that year. Therefore, we are unfortunately unable to establish more recent analytical results. However, we have discussed the potential current and future state of EFE violations based on studies with more recent and projected data in Section 4.1. We will make our reasoning for the time periods clearer in the revised manuscript.

Regarding the Referees' last comment on hydrological record selection, we are unsure about its precise aim, but interpret this to consider the selection of observed data. As we did not include observed data in the manuscript, we are unfortunately unable to address this point. However, all of the models selected for the analysis have undergone model validations against observed discharges as we note in AC 2.4.3.

**RC 2.4.6.** li 153-155. You actually exclude a fairly significant number of basins - did you test what the implications might be? At the least you need to objectively acknowledge this fact and its implications, rather than word it in such a way as to downplay it.

**AC 2.4.6.** The reason for excluding these basins is that they are smaller than a GHM grid cell. The maximum size of a GHM grid cell is approximately 55 km x 55 km at the Equator. Combined, 352 excluded sub-basins would comprise approximately 1 million $km^2$ at a maximum (assuming they are all located at the Equator), which is less than 1% of the global land area. We will revisit this implication to take note of the excluded area in a more objective way.

**RC 2.4.7.** li 163 - The EF methods. The strengths and limitations of these methods are not provided, but need to be clear to the reader. Table 1 just lays out the methods. There are a number of more recent hydrologic methods that are less simplistic and/or include a greater diversity of ecologically meaningful flow metrics. You have not used any of those methods in the composite approach. The reasons need to be given, especially given the assumptions made in the paper around ecological threat/safety. The paper would benefit from some discussion of why more sophisticated methods including regional approaches (ELOHA based) were not possible to apply at this time.

**AC 2.4.7.** The main reason for not using ELOHA based approaches is the massive investment of resources, which is out of reach in a global study like ours. Presumptive standards, such as those that we use in this work, have been proposed as "default placeholders" in cases where more advanced methodologies are out of reach (Richter et al., 2012). For further justification behind our selection of the EF methods and the ensemble approach, see AC 2.4.3. and AC 2.4.8. We hope that revising the introduction, especially according to AC 2.3.4., will argue for the selection of relatively simplistic methods in this global scale study. Further, we will reframe the parts of the Discussion in which we assess the ecological uncertainty of our method.

**RC 2.4.8.** Li 180-183. This section of text needs careful attention. "As the EFE lower bound, we selected the median of the EFR distribution…potentially consist of unrealistically low or high EFR estimates, caused by either highly deviant discharge provided by certain GCMs or distinctively different representation of ecosystem water needs in the EFR method". These two points need to be broken down and carefully argued. They are based on assumptions without any apparent ecological rationale. This needs to be clear, e.g. is the choice of median as the lower bound arbitrary? It designates the low-high flow boundary sometimes hydrologically, but that is not the same as a boundary designated for eflows purposes. Also, the last point on "distinctively different representation of ecosystem water needs" is hard to follow - what are you intending to say here?

**AC 2.4.8.** Selecting the EFE lower bound from the median of EFR distribution is an ensemble modelling approach, which is often used in multi-model studies incorporating global hydrological data (Peel and Blöschl, 2011; Zaherpour et al., 2018). As noted in AC 2.4.3., the individual EFR methods are indeed simplistic. Following the ensemble modelling approach, we choose to use the median estimate from a set of uncertain estimates. Taking the mean or median estimate from an ensemble of uncertain estimates has been shown to be effective regarding modelled vs. observed streamflow (e.g. Arsenault et al., 2015; Huang et al., 2017), which is why we choose to do it also for the EFRs.

We consider that as all of the selected EFRs are uncertain ecologically, it is better to select the lower bound following the ensemble paradigm. The other option would be to pursue an ecological best fit. This may not be possible due to the methods' simplicity and the lack of global-scale, observed data on ecosystem integrity, which would be needed to select the best fit. Moreover, the uncertainty of the underlying hydrological data is not resolved by only adopting more ecologically advanced methods. We will provide a more careful argument for the ensemble approach in the revised manuscript.

Regarding the "distinctively different representation of ecosystem water needs", the sentence aims to highlight that the methods define ecosystem water needs differently, and if some of the methods prove very different from others in certain contexts, those are excluded by taking the median. For example, Smakhtin's method has received criticism for its low EF allocation (Richter et al., 2012). Therefore, the median can be interpreted as a consensus estimate among the different EF methods. We will rephrase this sentence to clarify our argument.

**RC 2.4.9.** Li 207. "we excluded time periods during which the EFE is violated for less than three consecutive months"- by definition you would then exclude floods. Can that be correct?

**AC 2.4.9.** Here, we aimed not to analyse rapid, event-based floods but rather long-standing decreases or increases in discharge. Therefore, floods lasting less than three months are excluded; and floods lasting less than a month would also be excluded even if we did not enforce the three-month rule. However, we consider that enforcing the three-month rule is required for a robust interpretation of the results as outlined in AC 2.4.1. Furthermore, shorter floods themselves are essential for certain ecosystems (Hayes et al., 2018; Junk et al., 1989;

Schneider et al., 2017), but long-term increases in discharge may be harmful for them. If we considered short-term floods as violations, we would be going against the ecological foundations of the systems that require short-term floods. In the revised manuscript, we will more carefully elaborate on our rationale behind the three-month rule.

**RC 2.4.10.** li 213. " excluded sub–basins with extremely low flow from our analysis" This is a limitation that warrants more explanation or returning to in the discussion section. You also need to clarify the extent to which temporary rivers might be excluded as a result and to discuss the implications (see new Nature paper by Messager et al. 2021 on the global importance of such systems)

Messager, M.L., Lehner, B., Cockburn, C. et al. Global prevalence of non-perennial rivers and streams. Nature 594, 391–397 (2021). https://doi.org/10.1038/s41586-021-03565-5

**AC 2.4.10.** First, it should be noted that we determine the sub-basins by their outlets (lines 155-156). This means that for an exclusion, the outlet discharge in particular must be less than 10 $m^3$ $s^{-1}$. This, combined with the typical sizes of HydroBASINS level 5 (line 154), limits the exclusion to cover only extremely arid areas. This is shown in e.g. Fig. 2. where the excluded sub-basins are drawn with grey colour. In total, the number of excluded sub-basins during the recent past period is 522, covering an area of approximately 8.8 million $km^2$ (roughly 6.8% of the global land area).

However, as we simplify all upstream cells to contribute to the outlet cell, any upstream temporary rivers - as well as any violation status in upstream reaches - are masked. In the revised manuscript, we more explicitly discuss the scale dependency of our results and note that the exclusion discharge of 10 $m^3$ $s^{-1}$ must occur at the sub-basin outlet.

**RC 2.4.11.** li 214. You excluded basins if the mean flow was less than 10 cumecs – that is rather high as so many regions have the bulk of their rivers smaller than this. Interestingly the resolution of your maps does not show these smaller exclusions. By way of illustration – the Mgeni River in South Africa supports 6 million inhabitants in two cities Pietermaritzburg and Durban. The river just above the estuary in its natural condition only exceeded 10cumecs for 38 days a year, so this entire river should be excluded from your map – yet it does not show as excluded. This is concerning and should be discussed.

**AC 2.4.11.** As noted in AC 2.4.10., we consider only the sub-basin outlet discharge for the exclusion, notwithstanding the upstream river reach configuration. This is why many upstream areas drier than the threshold appear in the results.

Regarding the particular example of the Mgeni River, the HydroBASINS level 5 is too coarse to represent this as an individual basin as the catchment area of the Mgeni River is relatively small (approximately 4,400 $km^2$). Instead, the basin is lumped into a collection of coastal basins by the HydroBASINS data. This, however, causes many independent small basins to be represented by the highest-discharge cell of the lumped basins. In the revised manuscript, we will take note

of this special case where many coastal basins are lumped into one in HydroBASINS level 5, and implicate that higher detail - both in sub-basin catchment boundaries and discharge data - would be required for practical applications in these cases.

**RC 2.4.12.** li 225 - "classified each month of record into low (Q < 0.4MAF), intermediate (0.4MAF ≤ Q ≤ MAF), and high (Q > MAF) flow classes". On what basis? What evidence is here from the literature to support this categorisation or was it somewhat arbitrary?

**AC 2.4.12.** The categorisation to flow seasons is a synthesis from the selected EF methods presented in Table 1. Three out of five methods determine high flows to consist of months where MMF > MAF. Those two that define intermediate flow months determine low flow months to consist of months where MMF < 0.4 x MAF, which we select as the low flow boundary to incorporate intermediate months also. Although the precise class limits could be debatable, the main aim of separating the results to flow seasons is to illustrate that there are large differences in EFE violations depending on the amount of discharge. We will clarify the justification and the aim of separating the months to flow seasons in the revised Methods section.

**RC 2.4.13.** li 234. Statistical procedures require supporting references. e.g. for Kendall rank correlation coefficient.

**AC 2.4.13.** We will add a reference (Hollander and Wolfe, 1973) to support the Kendall rank correlation coefficient and another (Chambers et al., 1990) to support the linear regression slope t-test.

**RC 2.4.14.** li 236 - "Finally, we combined the EFE violation frequency and severity throughout the recent past time series with the linear violation trend slopes and performed a fuzzy c–means clustering (Bezdek, 1981) to each flow season separately." This sentence needs breaking down into clear points. Reasons need to be given for choices made e.g. for fuzzy c-means clustering. This is important. e.g., as Fig 5 relies heavily on the clustering method adopted.

**AC 2.4.14.** We will break the sentence into shorter parts. Further, we will provide additional reasoning for the clustering method as this was something pointed out by the third Referee (see AC 1.4.6.).

**RC 2.5. Results**

**RC 2.5.1.** li 239 - "EFE violations are widespread around the world, concentrating on lower bound violations in the arid and dry temperate climate zones" The pre-alteration context needs to be given, for comparison and for these results to have meaning. Also, there are limitations of the approach for rivers with very low to no flows, and this needs covering in the results and discussion. Again, there are issues around grammar and the expression of information that detract from the sense of this sentence (violations that concentrate on lower bound violations). While I understand what is intended, such sentences need to be corrected.

**AC 2.5.1.** Regarding the limitations of the approach, we hope that revising the Methods section, especially considering AC 2.4.1. and AC 2.4.7. to AC 2.4.10., the results will prove easier to interpret and discuss later.

As outlined in AC 2.4.1., we performed a new analysis on the frequency of pre-industrial EFE violations. Using the method to produce Figure 2, we get 24.0% of basins with more than 5% of all months 1801-1860 violated, vs. 49.8% of basins during 1976-2005. It is noteworthy that no sub-basins have more than 5% of months violated solely due to upper bound violations. Hence, all major upper bound violations are due to changes in discharge that have occurred between the pre-industrial and recent past periods, whereas a part of the major lower bound violations are caused by natural variability.

Given the 5% threshold, the global EFE violation frequency has doubled since the pre-industrial time period, which already indicates changes in discharge. This is increasingly highlighted when counting sub-basins with more than 10% (9.6% pre-industrial vs. 32.7% recent past) or more than 25% (0.08% pre-industrial vs. 9.5% recent past) of all months violated. We will incorporate these results in the revised manuscript and discuss them accordingly. We have also attached here a map (Fig. 1), showing the change in EFE violation frequency from the pre-industrial period to the recent past period. This will replace Fig. 2 in the manuscript. Further, we will also include maps of the pre-industrial violation frequency as part of a new Figure A4.

[Figure]

**Figure 1:** Frequency of environmental flow envelope (EFE) violations between 1976 and 2005 for both upper and lower bounds (a), lower bound only (b), and upper bound only (c) aligned with the change in violation frequency since the pre-industrial (1801-1860) period (d-f). All values are computed across four global hydrological models (GHMs). Sub–basins with mean annual flow (MAF) less than 10 m³ s⁻¹ at the sub-basin outlet are excluded.

**RC 2.5.2.** li 245. "Therefore, the EFE is rather violated by insufficient than excessive discharge, and regional patterns are more clearly visible in EFE lower bound violations whereas EFE upper bound violations are more dispersed into individual sub–basins." I think the results need breaking down further to make this argument. How much of this pattern is a result of: using hydrology based approaches that tend to focus more on the low flow regime (Q90) or monthly time step, and/or less on event-based influences such as floods; or a regional effect attributable to naturally more arid conditions rather than a function of the operation of infrastructure? How

sensitive are the results to the somewhat arbitrary setting of the upper and lower bounds, and to natural flow regime characteristics?

**AC 2.5.2.** Overall, wider areas of sub-basins are covered by EFE lower bound violations, which is why we consider it more likely than EFE upper bound violations. On the other hand, sub-basins with EFE upper bound violations are seldom grouped together as are sub-basins with EFE lower bound violations.

We hope that our extended justification of the method, especially regarding 1) the setting of lower and upper bounds (AC 2.4.8. & AC 1.2.1.), 2) the enforcement of the three-month rule (AC 2.4.1. & AC 1.3.3.) and 3) the role of floods versus long-term increases in discharge (AC 2.4.9.) are adequate to better prepare the reader for the Results section.

After we have also quantified the pre-industrial violations to investigate how large a fraction of the violations could be natural, there is strong evidence that the violation frequency has increased globally since the pre-industrial time. Regarding the comparison "rather violated by insufficient than excessive discharge", however, we do agree with the Referees that this might not be a feasible statement due to the different methods of defining the EFE bounds. If e.g. the EFE upper bound was defined with a lower percentile than the 95$^{th}$, the comparison could look very different. We will temper this argument in the revised manuscript.

**RC 2.5.3.** li 273 - Niger River - for a named river, the R should be upper case. Tigris-E Basin? Northern China - the N might need to be lower case unless it is a specific region.

**AC 2.5.3.** We will make sure that the naming of rivers and basins is correct throughout the manuscript.

**RC 2.5.4.** li 259-276. The results are interesting and some of them could feature more substantively in the discussion.

**AC 2.5.4.** We are happy to see these results being of interest. We will feature them more substantively in the discussion.

**RC 2.5.5.** li 287 - "Respectively…"Such sentences are poorly structured and compound, and need to be rewritten for clarity.

**AC 2.5.5.** We will rewrite these sentences for a clearer expression.

**RC 2.5.6.** li 300 - "increasing discharge alleviating EFE lower bound violations may turn out to be amplifying for EFE upper bound violations in some regions and downplay the positive indications of decreasing EFE lower bound violation trends." Such sentences cannot be understood as written.

**AC 2.5.6.** We will split and reword this sentence to make it understandable.

**RC 2.6. Discussion**

**RC 2.6.1.** The discussion needs greater depth in content, including in relation to ecological limitations, and the implications of the results. Draw in some points on more arid river systems (where the approach seems particularly weak) and on temporary rivers.

**AC 2.6.1.** Especially relating to comments AC 2.4.1. and AC 2.4.7. to AC 2.4.10., we hope that changes to previous sections, especially Methods, help in interpreting the discussion with respect to the limitations and implications of the study. We will expand and clarify our arguments, as well as address the limitations of the study in more detail.

**RC 2.6.2.** li 346 - Suggest you differentiate between ecosystem change and threat - e.g. in Europe many rivers are already ecologically degraded, not simply at threat of degradation.

**AC 2.6.2.** We will make this distinction by noting that some European rivers are indeed degraded and not only at the threat of degradation (Grizzetti et al., 2017). This is one example of a "threat" that has already been realised (see also AC 2.1.3.).

**RC 2.6.3.** li 348 - "Since these areas show relatively little EFE violations, it can be inferred that even though the quantitative discharge would be within the EFE, the anthropogenic flow alteration can still be major." Should read "relatively few/minor". This is a key point that needs more elaboration - otherwise, it suggests that the approach taken/methods used are insufficiently ecologically representative or attuned to reality.

**AC 2.6.3.** We will combine this observation with the revised section on ecology and elaborate it with respect to the extended description of Methods.

**RC 2.6.4.** li 359 - "can largely be attributed to anthropogenic impact…" But one might argue that you simply see potential congruency rather than having demonstrated an attribution/causal link through the data?

**AC 2.6.4.** We will reword the sentence so that it doesn't imply a causal link but rather a congruency.

**RC 2.6.5.** li 362 - What is meant by "remotely teleconnected" here?

**AC 2.6.5.** The "remotely teleconnected" refers to changes in the hydrological cycle, which have an effect within a sub-basin despite originating to regions outside its boundaries. For example, land use change can alter atmospheric moisture recycling and affect e.g. rainfall on distant sub-basins, which are not connected to the origin of the change by land surface hydrology. We will clarify this in the revised manuscript.

**RC 2.6.6.** Li 366. This is suggesting that drought is the main cause of increasing violations – not increased abstraction? That is strange as demands in these regions are high.

**AC 2.6.6.** We aimed to state that droughts and EFE violations co-occur in many regions and the underlying driver - an abnormally low amount of water in river systems - is the same. Based on the available evidence, we are unable and have not attempted to establish causal links. We will revise the sentence to avoid either claiming that drought causes the violations, or vice versa.

**RC 2.6.7.** li 384 - "flaws" - suggest reword as "limitations" and ref. Poff et al. 2017 here. If they are flawed, then this puts their use into question.

**AC 2.6.7.** We will reword "flaws" to "limitations" and incorporate the suggested reference.

**RC 2.6.8.** li 384 - "global fish biodiversity has shown that several other factors, such as water quality and the presence of invasive species, may be more important in maintaining riverine ecosystems than quantitative flow (Su et al., 2021)." Yes, but the context is important to get right here and in general. Moreover, fish are one group of species, not directly equivalent to an ecosystem or its functions. This section on ecology is weak and needs a very careful read-through to address assumptions/oversimplifications.

**AC 2.6.8.** We will revisit the section to more explicitly cover the assumptions and limitations of our method, in accordance with AC 2.4.7. to AC 2.4.10. Regarding the referenced study, we chose it to be referenced here because it is a recent global study based on *in situ* measurements. However, we do agree that fish are not equivalent to the complete ecosystem and will emphasise this in the revised sentence.

**RC 2.6.9.** li 386 - "well-being" is typically used for people, and integrity or health for ecosystems.

**AC 2.6.9.** We will replace "ecosystem well-being" with "ecosystem integrity" throughout the text.

**RC 2.6.10.** li 394-404. This is a mix of natural flow paradigm, novel and hybrid systems, designer flows, shifting baselines etc. and needs to be sorted out into a more cogent argument.

**AC 2.6.10.** The aim of this paragraph is to first state that EFEs are based on the natural flow paradigm, which is widely adopted in the broader context of quantifying human impacts on the Earth system. However, if EFEs were to be applied in practical terms, following the simple aim of retaining or returning to the natural state is not straightforward. This is because of the profound anthropogenic modification of rivers, as well as climate change that affects even pristine streamflow. Therefore, any measure should recognise the practical limits of flow restoration. We will clarify and simplify the argument to convey our message better. We will also incorporate the reference suggested in AC 2.2.1. (Poff et al., 2016) to give an example on the practical process of allocating streamflow to human use and the environment.

**RC 2.6.11.** li 396 - "its globally equal absoluteness". This is not English as written.

**AC 2.6.11.** We will revise the sentence to a better wording.

**RC 2.6.12.** li 408 - Section 4.3 needs a serious edit. Please use simpler sentences that make clear points, to help with the logic of your argument.

**AC 2.6.12.** We will revisit the Section to convey our argument in a more effective way. Here, we attempted to first compare our method with relevant literature and then evaluate the limitations of the method. Especially regarding the limitations of the method, we hope that changes according to AC 2.4.7. to AC 2.4.10. will help in clarifying this part.

**RC 2.6.13.** li 409 - You need to provide a reference or better introduce "ensemble thinking" in this context - here or under the methods.

**AC 2.6.13** We will add a better introduction to "ensemble thinking" in the Methods section; see AC 2.4.8.

**RC 2.6.14.** li 427. Terms such as "excessive flows" need to be expressed using more scientifically appropriate wording.

**AC 2.6.14.** We will reword "excessive flows" to "increased upper extreme flows" throughout the text.

**RC 2.6.15.** li 447 - "quantification of the riverine ecosystem responses to prolonged and excessive flows through case studies would benefit the development of the EFE upper bound." Suggest include some supporting references here for a couple of e-flow case studies that address this aspect.

**AC 2.6.15.** Although there are studies to support the claim that long-term increased flows are harmful for certain ecosystems (Hayes et al., 2018; Junk et al., 1989; Schneider et al., 2017), we are currently unaware of quantitative case studies that would determine the upper bound. Here, our aim was to highlight that the quantifications in particular are missing, and future studies could aid in replacing our EFE upper bound with a more ecology-based alternative. We will revise the sentence to make our point clearer.

**RC 2.6.16.** Li 455 – it would be useful to comment on the application of this approach at the local scale compared to the global scale, and the limitations associated with that. In line 466 you mention "operationalising our results at the basin scale", but just what this means could be clarified.

**AC 2.6.16.** We hope that the revised Methods section, as well as the added discussion on the limitations of this method prepares better for the end of the Discussion. Using the coarse simulated data at the level 5 HydroBASINS sub-basins at a monthly timestep, the EFE, as we define it, is best used in relatively large sub-basins with a clear outlet. For more local applications, especially regarding small coastal catchments (e.g. the Mgeni river), EFEs based on global data may not be applicable, although the EFE as a concept would be. The need for

more sophisticated EF methods, as well as the consideration of catchment-specific characteristics such as flow intermittency, are increasingly highlighted when transferring the approach into higher-detailed sub-basins. In the revised manuscript, we will provide a more careful argument on the scale dependency of our results.

**RC 2.6.17.** li 466. Can you be more specific on ways to make the approach more meaningful operationally i.e. from an implementation standpoint? There are obvious limitations, given the characteristics of the EF methods used. So the appropriateness and the limits of such simple global scale, hydrology- based methods must be kept in mind in these concluding remarks.

**AC 2.6.17.** After revising the text according to AC 2.6.16., we hope that the assumptions and limitations of our method are clearer at this point. As in many other global-scale studies, the best way to make use of our results is to preliminarily identify areas in which the most drastic EFE violations prevail. Our in-depth analysis of the violation characteristics performs this well, but basin-scale decision-making should consider higher-resolution data and more advanced EF methods due to the multiple limitations outlined in these responses. It should also be kept in mind that our method may not comprehensively portray the ecosystem conditions due to its simplicity, and locally relevant violations can be masked by the data and method resolution. Moreover, our results are best used in larger catchments instead of smaller ones covering few grid cells (see AC 2.4.11.). We will revisit the Discussion and Conclusion to more explicitly feature the limitations and implications of using solely global data and hydrological EF methods.

**RC 2.7. Figures and tables**

The paper follows a nice, clear and logical layout and the figures are generally good.

**RC 2.7.1.** Fig 3. It is difficult with these colours to differentiate the low violations cases from the less than 10% frequency case. Also, is it well explained in the text as to what is meant by the shorthand "no months in season"?

**AC 2.7.1.** The "no months in season" refers to sub-basins in which there are no months between 1976 and 2005 during which monthly discharge would fall below 0.4 * MAF. We will add a mention of this in the figure caption. Regarding the colours, we interpret the Referee to point out to the distinction between "no violations" blue colour and the class of < 10% frequency & low severity, which is represented by a grey colour. We will revise the blue colour to be more distinctive from the grey.

**RC 2.7.2.** Fig A1 - There is too much small text in the figure.

**AC 2.7.2.** Thanks for the note; we will revise some of the text into a more concise form to make the figure less crammed.

**Recommendation**

Moderate revision.

We are grateful for the time taken and effort made by the Referees to provide their comments on the manuscript. We hope that our responses are adequate and revising the manuscript according to them will make it significantly stronger.

---

## Author Response (AR2)

**Report #1**

Widespread and increasing violations of environmental flow envelopes

**General comments**

Since the last revision, the authors have answered and addressed the most important comments of the reviewers. Especially the inclusion of pre-industrial violation results and the omission of overly strong language has markedly improved the manuscript.

However, there is still need for further clarification and specification in some aspects of the abstract and discussion. By improving these sections this study could be presented more robust. Moreover, there is also an opportunity to improve the writing. Therefore, I recommend minor revisions.

We thank the reviewer for their encouragement, as well as for the further comments and suggestions. We are also grateful for the specific and insightful suggestions on how to improve our writing, which certainly help us to better communicate our results. We hope that our minor revisions and clarifications are adequate in order to improve the robustness and style of the manuscript.

**Specific comments**

Note that some of the comments to do with the writing are based on my personal writing experience, and are not "wrong". However, they may help the authors to improve their writing throughout the document.

Line 28: What uncertainty? Please clarify

This uncertainty is related to using a limited number of both discharge data sets and EF methods, and we have now specified this here. Lines 27–29 now read as follows:

*"Environmental flows (EFs) have emerged as a prominent tool for safeguarding the riverine ecosystems, but at the global scale, the assessment of EFs is associated with high uncertainty related to the hydrological data and EF methods employed."*

Line 29: "The sub-basin specific EFE is" to "Sub-basin specific EFEs are"

We have reworded this; lines 30–31 now read as follows:

*"Sub-basin specific EFEs are determined for approximately 4,400 sub-basins at a monthly time resolution, and their derivation considers the methodological uncertainties related with global-scale EF studies."*

Line 30: "and its derivation considers the methodological uncertainties", again please clarify

We hope that the change to the previous sentence (comment on line 28) in lines 27–29 will make it clearer that this "methodological uncertainty" relates to the hydrological data and EF methods.

Line 32: Maybe mention Q95 here?

In the abstract, we would like to refrain from explicitly mentioning how the EFE upper bound is determined, as it would unnecessarily complicate the abstract with technical definitions. To keep it less technical, we haven't provided the technical definition of the EFE lower bound, either.

Line 32: "This" to "This upper bound"

We have reworded this; lines 32–33 now read as follows:

*"This upper bound enables identifying areas where streamflow has substantially increased above natural levels."*

Line 33: "Long-term" is used throughout the manuscript, for different meanings. This is sometimes confusing. See also some comments below. "Long-term" to "annual"?

We agree with the reviewer that the use of "long-term" was sometimes confusing, and we have replaced all instances related to "long-term flow alterations/EFE violations" with "persistent flow alterations/EFE violations". Here, our intention was to illustrate that commonly, EF studies show violations for one period of time. For example, Jägermeyr et al. (2017) report EF deficits averaged over 1980–2009, whereas we complement this by assessing the trends within 1976–2005.

Line 37: Best to move "global hydrological model outputs from the ISIMIP2b ensemble" to its own sentence (perhaps at the start of the paragraph "Discharge was derived from ...").

We have reworded this; lines 37–39 now read as follows:

*"Discharge was derived from global hydrological model outputs from the ISIMIP 2b ensemble. We use pre-industrial (1801–1860) quasi-natural discharge together with a suite of hydrological EF methods to estimate the EFEs. We then compare the EFEs to recent historical (1976–2005) discharge to assess the violations of the EFE."*

Line 40: "widespread, occurring" to "widespread and occurring", as one refers to the spatial extent and the other the temporal extent.

*We have reworded this; lines 41–43 now read as follows:*

*"The EFE violations are widespread and occurring in half of the sub-basins of the world during more than 5% of the months between 1976 and 2005, which is double compared to the pre-industrial period."*

Line 44: "spatially distributed" to "dispersed" as in the discussion.

*We have reworded this; lines 44–45 now read as follows:*

*"Indications of increased upper extreme streamflow through EFE upper bound violations are relatively scarce and dispersed."*

Line 46: Ending on a positive note would be stronger, what is the broader application of your study?

*We have reorganised this sentence to end on a positive note and outlined that the globally robust EFEs can inform global research and policies on water resources management. Lines 45–48 now read as follows:*

*"Although local fine-tuning is necessary for practical applications, and further research on the coupling between quantitative discharge and riverine ecosystem responses at the global scale is required, the EFEs provide a quick and globally robust way of determining environmental flow allocations at the sub-basin scale to inform global research and policies on water resources management."*

Line 54: "population growth", not only growth but also agricultural and population development (irrigation expansion and diet changes).

*We have added agriculture (especially irrigation water use) as an additional factor contributing to the pressure on freshwater ecosystems, as agriculture is responsible for a major share of all water use, and future unsustainable water consumption is projected to increase (Campbell et al., 2017; Wada and Bierkens, 2014). Lines 56–58 now read as follows:*

*"The pressure on freshwater ecosystems is only expected to increase in the future due to population growth, agriculture (especially irrigation water use), and projected climate change (Best, 2019; Campbell et al., 2017; Graham et al., 2020; Thompson et al., 2021, Wada and Bierkens, 2014)."*

Line 63: Probably remove "in addition" as it specifies, and not adds to, the previous sentence?

We have reworded this; lines 66–67 now read as follows:

*"Human actions impact the intra- and interannual variability, which are often considered as parts of the natural flow regime (Richter et al., 2006)."*

Line 124: What are "mechanistic equations"? Probably use "process-based"?

We have reworded this; lines 126–128 now read as follows:

*"Simulating discharge with the GHMs involves modelling the global terrestrial hydrological cycle through process-based equations, as well as forcing the models with observed or modelled climate."*

Figure 1: Could the three month threshold be included here?

We have now included the three-month threshold in Figure 1.

Line 165: What do these levels mean? Maybe omit?

The intention of including the HydroBASINS scale levels in the text was to show that we used a medium detailed sub-basin division in this study, as we were unable to use the highest detail levels, but that we included an adequate level of detail in our analysis. We have now revised the beginning of this paragraph to make this point clearer, and lines 166–171 now read as follows:

*"We used the HydroBASINS sub-basin division, which is a global polygon layer series dividing the world into sub-basins at different scale levels from the lowest detailed level 1 to the highest detailed level 12; we selected the medium detailed level 5 (Lehner and Grill, 2013). Within each level, the geographical areas of sub-basins are relatively equal, and level 5 is the highest level of detail that can be rasterized into a 0.5-degree resolution grid without an excessive loss of sub-basins that are smaller than a grid cell."*

Line 200: Normal distribution cannot be assumed for discharge. I did a quick test for 100 of the world's largest rivers, and normality could only be assumed for half of them (using the Shapiro-Wilk test on multi-year average measured discharge). Moreover, why is it assumed there are errorous outliers in model simulations that should be removed, how do simulations make errors? This also affects the Q95 upper bound in the study. If so, this should be discussed.

We agree with the reviewer that assuming normal distribution of discharge globally is unsubstantiated, and we have omitted the claim potentially implying so. Here, we did not implicate anything related to the cause of outliers since they can also be extreme natural events (e.g. thousand-year floods). However, we consider that removing outliers, which could largely affect the computation of EFRs and shift the EFE upper bound very high, is justified also considering that some extremely rare natural events may be excluded due to this. If the extreme

outliers were left in the data, the EFE upper bound may rise very high based on extremely rare conditions and potentially mask some of the current EFE upper bound violations. We have revisited the text related to the outlier removal, and lines 204–208 now read as follows:

*"Before computing EFRs, we removed monthly outlier discharge further than three standard deviations away from mean monthly discharge. This procedure only removed extremely deviating discharge values, which could greatly distort the computation of EFRs or shift the EFE upper bound very high if left in the data. Similarly for the resulting EFR distribution, EFRs further than three standard deviations away from mean EFR were removed. This way, we avoided skewing the EFR distribution with extreme outliers in pre-industrial data."*

Furthermore, in Section 4.4 lines 518–520, we've added a sentence outlining that although useful for eliminating potential errors, the outlier removal is also a limitation of the study as it might exclude the most extreme natural events:

*"Moreover, to prevent raising the EFE upper bound extremely high, we excluded outlier discharge prior to determining the EFEs (Sect. 2.2), which may result in excluding not only potential model errors but also extremely rare natural events. "*

Line 328: "figures" to "values" or "results".

We have reworded this; lines 333–334 now read as follows:

*"These values mean that the typical EFE lower bound violation is caused by discharge falling 19–37% below the EFE lower bound."*

Line 423: "both long-term and recent". "Long-term" here is referring to "more than three consecutive months". However, as long-term is used in combination with "recent", which refers to the last 30 years, which is confusing. As an alternative to using "long-term" I would propose to use "extended" or "persisted" throughout the manuscript?

We have revised this terminology according to the comment on line 33, and here lines 425–427 now read as follows:

*"Given that the change from the pre-industrial period is substantial (Fig. 2d) and all considered violations last three or more months (Sect. 2.3), the EFE violations represent persistent flow alterations during the recent historical period."*

Line 425: If possible, could the absolute values be included in the supplementary?

We will release the R code and data used to compose the results in an open repository. In addition, we will include the results underlying Figure 3 in the supplementary.

Line 434: What is a possible reason for this wider spread? Is it related to the model ensemble?

Methodological differences between our study and Jägermeyr et al. (2017) can be assumed to be a possible reason for the wider spread. Mainly, the differences stem from time periods used in determining the "natural" discharge (1801–1860 vs. 1980–2009), as well as the number (five vs. three) and aggregation (median vs. mean) of the EF methods. In addition, we include a greater number of global hydrological models (GHM) forced with outputs from general circulation models while Jägermeyr et al. (2017) use one GHM (LPJmL) forced with observed climate. In our LPJmL-specific results (Figure S4), the EFE violations are spread similarly, agreeing with other models of our analysis. Hence, the difference in time periods and EF methods could be assumed to be the main cause of differences. In the revised manuscript, we have moved the sentence outlining the methodological differences between the two studies right after introducing the differences in results, and lines 439–442 now read as follows:

*"Our EFE violations are more widespread in large parts of Australia, South America, and Southern Africa (Fig. 2–3) than those reported by Jägermeyr et al. (2017). However, Jägermeyr et al. (2017) determine EFRs based on pristine discharge simulation between 1980 and 2009 and report annual averages, which differs from our seasonal analysis that is based on the pre-industrial period and includes potential climate change impacts."*

Line 437: Can you expand upon these limitation (perhaps with citations)? Or maybe move this to section 4.3, where these limitations are discussed.

We have moved this statement to Section 4.3 in which the limitations are discussed in more depth.

Line 441: "(i.e. not taking potential climate change impacts into account as we do)" would be better at the end of the sentence, were you discuss what your study does. "(...) which is different from our seasonal analysis based on the pre-industrial period and includes potential climate-change impacts"

We have moved the sentence including this phrase upwards and revisited it (see previous comment on line 434).

Line 451: "cautious interpretation of our results" to "cautious interpretation of our results in these regions"

We have reworded this; lines 454–456 now read as follows:

*"Regarding the Pan-Arctic areas in particular, GHMs have recently been shown to perform relatively poorly (Gädeke et al., 2020), which calls for cautious interpretation of our results in these regions."*

Line 465: Would omit this sentence as it is discussed before and is addressed in the results.

We agree with the reviewer and have omitted this sentence.

Line 491: Although the limitations of EFs are described, could you indicate how they are useful? Currently this section seems to undermine the value of your study.

As mentioned in Section 4.4, the benefit of using simplistic EF methods is that they can provide a globally consistent overview of anthropogenic flow alteration. We have revisited this sentence to be more balanced and to highlight the benefits of the simplistic methods, and lines 498–501 now read as follows:

*"For practical use and analyses of sub-basin scale riverine ecosystem integrity, our results should be complemented by local studies using holistic EF methods. However, our globally consistent approach using hydrological EF methods provides a comprehensive global overview on anthropogenic flow alteration."*

Line 509: Important for what? Please elaborate.

Here, we aimed to state that temporary rivers are common and important for the local ecosystems supported by them. We have reworded this statement, and lines 523–525 now read as follows:

*"This is a notable limitation particularly in the case of temporary rivers, which have recently been shown to comprise a large part of global rivers, and which are highly important for local ecosystems (Messager et al., 2021)."*

Line 513: Why this specific number? Maybe just omit?

Here, the intention was to clarify that the spatial resolution of our gridded discharge is relatively coarse, and relying on a small number of cells to determine and assess EFEs is highly uncertain. However, providing explicit numbers is ambiguous, and we have therefore reworded this; lines 528–529 now read as follows:

*"Applications at the scale of small catchments consisting of few 0.5-degree grid cells should rather resort to high-detail observed data instead of global data simulated in a coarse grid."*

Line 507: "On the one hand (...) but on the other hand". These sentences do not contradict, just use "and".

We have revisited this sentence along with the previous and the next comment.

Line 508: Is there also an advantage to the spatial aggregation (as there is for the temporal aggregation)? If so it would be good to mention here.

Using coarse grid scale could potentially provide unstable results due to high GHM variability in headwater and low-discharge streams. When we simplify the sub-basins to be represented by the outlet cell only, we mask the greatest upstream variability and increase the robustness of our sub-basin scale results. This draws attention to the sub-basin scale instead of showing potentially very uncertain and highly deviating cell-wise results. We have now made it more explicit that this is an advantage, and lines 520–523 now read as follows:

*"Spatially, we consider the sub-basin outlet location as representative for the whole upstream area, which simplifies the sub-basin into one hydrological unit (Sect 2.1). Using the coarse grid scale could potentially provide unstable results due to high GHM variability in headwater and low-discharge streams, which is countered by this aggregation. However, the aggregation also masks local EFE violations that may vary within the sub-basin itself."*

Line 521: Could you elaborate on why the separation of natural and anthropogenic flow alterations would give additional information on the "seriousness" of major violations?

The main reason why this separation would be useful lies in the measures to alleviate major violations. For example, if anthropogenic flow alterations (e.g. water withdrawals) were the main cause of EFE violations in a given region, the measures to alleviate the violations would be very different from those regions in which the main cause is climatic. We have added a sentence in lines 536–538 to elaborate on this claim:

*"Separating natural and anthropogenic flow alterations could prove useful in estimating how major violations – and in which regions – should be deemed as the most serious. This way, the actions to alleviate EFE violations could be best prioritised and targeted to the most affected regions."*

Line 530: "On one hand (...) On the other hand". These sentences do not seem to contradict. Maybe use "nevertheless" instead of "on the other hand" (and omit "on the one hand").

We have reworded this; lines 547–550 now read as follows:

*"Operationalising our results at the basin scale requires more detailed data, assimilation of cross-scale information, and interdisciplinary knowledge to more fully portray the ecological and hydrological conditions of each unique river. Nevertheless, our results highlight the need to consider environmental flows in both global research and policies on water resources management as major anthropogenic flow alterations prevail across wide areas."*

**Report #2**

**General comments**

This reviewer was party to a submission of considerable detail in the first round of reviews. To their credit the authors have clearly attended to most if not all of the comments previously given, which has considerably upgraded the paper. It is thus recommended for publication with minor edits as detailed below.

A major finding of the first review was that the absence of ecological consideration in this hydrological approach was not acknowledged. This has been partially rectified in a clumsy way and deficiencies remain, some of which are indicated below.

We thank the reviewer for their comments on the revised manuscript and agree that addressing the considerably detailed comments improved the initial manuscript substantially. We hope that our further clarifications and revisions will prove sufficient.

**Specific comments**

Title –"Widespread and increasing violations of environmental flow envelopes" – this heading does not convey the depth and substance of the paper particularly as the concept of flow envelopes is not immediately intuitive. At least the word "global" should be included.

We agree with the reviewer that especially omitting the word "global" from the title may undermine the value of the study. We have discussed among all authors about the title of the article, and agreed to change it to *"Globally widespread and increasing violations of environmental flow envelopes".* As discussed in the major revision, we'd be hesitant to include anything related to ecology in the title since we haven't done any explicit validation for the relationship between EFEs and ecosystem responses.

175 – quasi natural – while dams may not have been prevalent, canals were common enough in a few areas e.g. in England the construction of canals began in the mid-1700s. This would have impacted streamflow but only in selected areas.

We agree that minor anthropogenic modification of rivers may have already existed by 1860, however, e.g. large-scale hydropower dams and extensive irrigation schemes followed only after the industrial revolution. We have revisited this statement to be less absolute, and lines 178–182 now read as follows:

*"We defined the EFEs based on the pre-industrial period (1801–1860). While some flow alteration (e.g. canals) may have already existed by 1860, large-scale human modification of rivers has prevailed mainly after the pre-industrial period. For example, area equipped for irrigation has increased sixfold since 1900 (Siebert et al., 2015), and many of the globally*

*largest dams have been commissioned during the 20th century (Lehner et al., 2011). Therefore, we presumed that this time period is quasi-natural – i.e. near the natural flow regime."*

250 – it would be good to include at least a sentence of summary of the supplementary material.

We have added a sentence explaining the effect of changing the minimum streak length in lines 251–255:

*"In addition to results presented in the following section with a minimum three-month sequence of violations, we repeated the analysis with other minimum lengths of the violation streak. The results of this sensitivity analysis are presented in the supplementary material (Fig. S1–S3); shorter minimum violation streaks extend the violations to wider areas, and increasing the minimum violation streak limits the violations to relatively small regions."*

480 - these purely hydrological methods aim to establish the hydrological conditions that would be acceptable to ecosystems, but do not include any metric whereby this may be tested. Suggest this perspective be included.

We have added this perspective to the sentence, and lines 482–485 now read as follows:

*"Our method – and EFs in general – assumes that violating or respecting the EFE is associated with degrading or preserving riverine ecosystems. However, this might not hold for simplified hydrological EF methods as they lack metrics of assessing the correlation between presumably adequate hydrological conditions and ecosystem responses (Poff and Zimmerman, 2010; Richter, 2010; Richter et al., 2012; Mohan et al., 2021)."*

484 – "This is because the ecosystem response to altered flow regimes varies across spatial and temporal scales, as well as between different species" – this sentence does not cover the issue so suggest mentioning that altered flows affect a range of ecological characteristics from sediments, to stream morphology including riparian banks, to biodiversity and community dynamics of most fauna and flora. Any of the reviews of EF will spell this out.

Regarding the impacts of flow alteration on ecosystems, we have added more detail according to the suggestion, and a reference to Poff and Zimmerman (2010). Lines 487–489 now read as follows:

*"This is because the ecosystem response to altered flow regimes varies across spatiotemporal scales and different species due to the impact of altered flow regimes on e.g. sediment transport, stream and riparian bank morphology, and community dynamics of fauna and flora (Biggs et al., 2005; Poff and Zimmerman, 2010; Poff et al., 1997; Rolls et al., 2018)."*

487 – "Though fish make up only a part of a riverine ecosystem, these studies support incorporating water quality–related factors in a comprehensive EF definition." It is not clear why water quality is given prominence here – when in most EF studies it is the response indicators

(biota) that are most important as they are the best indicators of the success of the EF. It is also complex to include WQ in any comprehensive way in determination of EF because the presence of WQ issues may be and generally is completely non flow-related.

We agree with the reviewer that water quality is not the only factor missing from comprehensive EF determinations, but water quality was the main determinant considered in the referred studies. Our intention was to state that many factors beyond discharge could benefit the determination of EFs, and we have now revisited the paragraph to less prominently focus on water quality only. Lines 493–498 now read as follows:

*"Recently, quantitative water flows have been shown to be less important than water quality and invasive species for assessing rivers' ecological status, determining fish biodiversity, and driving fish habitat loss (Barbarossa et al., 2021; Grizzetti et al., 2017; Su et al., 2021). Though fish make up only a part of a riverine ecosystem, these findings underline that discharge alone cannot provide a comprehensive EF definition but other factors should be considered, as well. Holistic EF methods that include these factors – and also observation of biotic responses – correlate much better with ecosystem states, but require in situ data, ancillary variables, and local expert knowledge (Poff et al., 2017; Tharme, 2003) that are not available at the global scale."*

494 – floodplains - this seems a strange example, because from an ecosystem point of view you would hardly call the necessary flooding of a floodplain a violation of the environmental flows.

We agree with the reviewer that necessary flooding of a floodplain should not be considered as an environmental flow violation. However, if the flooding occurs during a period in which the natural conditions are dry (EFE upper bound violation), the ecosystem dependent on distinct dry and wet periods will degrade (Hayes et al., 2018; Junk et al., 1989; Schneider et al., 2017). We have made the mechanism behind this example more explicit, and lines 504–506 now read as follows:

*"The link between EFE upper bound violations and ecosystems exists, since, for example, floodplain ecosystems in monsoon flood pulse systems require distinct dry and wet periods, and disturbing the dry period by increased discharge may degrade the ecosystems (Hayes et al., 2018; Junk et al., 1989; Schneider et al., 2017)."*

500 – " detrimental to riverine ecosystems outside monsoon regions" – it is not clear why there is the limitation of monsoon areas. Does this imply that they are detrimental to ecosystems inside monsoon areas?

As outlined in the previous comment, studies suggest that EFE upper bound violations during the naturally dry period will indeed be detrimental to ecosystems inside monsoon areas. Here, we have complemented the previous addition by making this sentence more explicit; lines 509–511 now read as follows:

*"Hence, EFE upper bound violations are strong signals of increased upper extreme flows, although it cannot be inferred from this study whether these are detrimental to riverine ecosystems beyond regions with distinct dry and wet periods, such as the monsoon areas."*

516 – "In the future, the EFEs should be developed by complementing our global analysis with more advanced EF methods and more detailed hydrological data that better correlate with riverine ecosystem status" – this is a presumptuous way of stating this, presumptuous in that it is your model that should form the basis of future EFEs

We agree with the reviewer that the sentence was formatted poorly, and we have revisited it to be less presumptuous. Lines 531–532 now read as follows:

*"In the future, global analysis with more advanced EF methods and more detailed hydrological data that better correlate with riverine ecosystem status could further develop the EFEs."*

**References**

Campbell, B., Beare, D., Bennett, E., Hall-Spencer, J., Ingram, J., Jaramillo, F., Ortiz, R., Ramankutty, N., Sayer, J., Shindell, D., 2017. Agriculture production as a major driver of the Earth system exceeding planetary boundaries. Ecology and Society 22. https://doi.org/10.5751/ES-09595-220408

Hayes, D.S., Brändle, J.M., Seliger, C., Zeiringer, B., Ferreira, T., Schmutz, S., 2018. Advancing towards functional environmental flows for temperate floodplain rivers. Science of The Total Environment 633, 1089–1104. https://doi.org/10.1016/j.scitotenv.2018.03.221

Jägermeyr, J., Pastor, A., Biemans, H., Gerten, D., 2017. Reconciling irrigated food production with environmental flows for Sustainable Development Goals implementation. Nature Communications 8, 15900. https://doi.org/10.1038/ncomms15900

Junk, W., Bayley, P., Sparks, R., 1989. The Flood Pulse Concept in River-Floodplain Systems. Canadian Special Publication of Fisheries and Aquatic Sciences 106, 110–127.

Poff, N.L., Zimmerman, J.K.H., 2010. Ecological responses to altered flow regimes: a literature review to inform the science and management of environmental flows. Freshwater Biology 55, 194–205. https://doi.org/10.1111/j.1365-2427.2009.02272.x

Schneider, C., Flörke, M., De Stefano, L., Petersen-Perlman, J.D., 2017. Hydrological threats to riparian wetlands of international importance – a global quantitative and qualitative analysis. Hydrology and Earth System Sciences 21, 2799–2815. https://doi.org/10.5194/hess-21-2799-2017

Wada, Y., Bierkens, M.F.P., 2014. Sustainability of global water use: past reconstruction and future projections. Environ. Res. Lett. 9, 104003. https://doi.org/10.1088/1748-9326/9/10/104003